# Exceptional parallelisms characterize the evolutionary transition to live birth in phrynosomatid lizards

Saúl F. Domínguez-Guerrero [1,2,3 ✉], Fausto R. Méndez-de la Cruz[2], Norma L. Manríquez-Morán [4], Mark E. Olson [2], Patricia Galina-Tessaro[5], Diego M. Arenas-Moreno [2,3], Adán Bautista- del Moral[2,3], Adriana Benítez-Villaseñor [2,3], Héctor Gadsden [6], Rafael A. Lara-Reséndiz [5,7], Carlos A. Maciel-Mata [4], Francisco J. Muñoz-Nolasco [2,3], Rufino Santos-Bibiano [2,3], Jorge H. Valdez-Villavicencio [8], Guillermo A. Woolrich-Piña [9] & Martha M. Muñoz[1]

Viviparity, an innovation enhancing maternal control over developing embryos, has evolved >150 times in vertebrates, and has been proposed as an adaptation to inhabit cold habitats. Yet, the behavioral, physiological, morphological, and life history features associated with live-bearing remain unclear. Here, we capitalize on repeated origins of viviparity in phrynosomatid lizards to tease apart the phenotypic patterns associated with this innovation. Using data from 125 species and phylogenetic approaches, we find that viviparous phrynosomatids repeatedly evolved a more cool-adjusted thermal physiology than their oviparous relatives. Through precise thermoregulatory behavior viviparous phrynosomatids are cool-adjusted even in warm environments, and oviparous phrynosomatids warm-adjusted even in cool environments. Convergent behavioral shifts in viviparous species reduce energetic demand during activity, which may help offset the costs of protracted gestation. Whereas dam and offspring body size are similar among both parity modes, annual fecundity repeatedly decreases in viviparous lineages. Thus, viviparity is associated with a lower energetic allocation into production. Together, our results indicate that oviparity and viviparity are on opposing ends of the fast-slow life history continuum in both warm and cool environments. In this sense, the 'cold climate hypothesis' fits into a broader range of energetic/life history trade-offs that influence transitions to viviparity.

[1] Department of Ecology and Evolutionary Biology, Yale University, New Haven, CT 06511, USA. [2] Instituto de Biología, Universidad Nacional Autónoma de México, 04510 Ciudad de México, México. [3] Posgrado en Ciencias Biológicas, Instituto de Biología, Universidad Nacional Autónoma de México, 04510 Ciudad de México, México. [4] Centro de Investigaciones Biológicas, Universidad Autónoma del Estado de Hidalgo, 42184 Mineral de la Reforma, Hidalgo, México. [5] Centro de Investigaciones Biológicas del Noroeste S. C., 23096 La Paz, Baja California Sur, México. [6] Instituto de Ecología, A. C., 61600 Pátzcuaro, Michoacán, México. [7] Facultad de Ciencias Exactas, Físicas y Naturales, Centro de Zoología Aplicada, Consejo Nacional de Investigaciones Científicas y Técnicas, Instituto de Diversidad y Ecología Animal, Universidad Nacional de Córdoba, Córdoba 5000, Argentina. [8] Conservación de Fauna del Noroeste, A.C., 22785 Ensenada, Baja California, México. [9] Tecnológico Nacional de México campus Zacapoaxtla, 73680 Zacapoaxtla, Puebla, México. ✉email: sauldguerrero@gmail.com

Live-bearing (viviparity) is a major evolutionary novelty in the tree of life that affords physiological control and protection to developing embryos, providing higher offspring survivorship[1–5]. There are at least 150 independent origins of viviparity in vertebrates, particularly in squamate reptiles (>115 origins)[6,7]. Given their strong representation in relatively cool habitats, live-bearing in squamates has been classically interpreted as an adaption to lower environmental temperatures (i.e., the 'cold-climate hypothesis', or CCH): relative to eggs deposited in nests, incubation within the dam provides a relatively stable (and warmer) temperature, in turn shortening developmental time and enhancing offspring viability in cool environments[3,8,9]. Complete embryogenesis within the female reproductive tract may also be accompanied by adjustments in life history, morphology, thermoregulatory behavior, and thermal physiology[10–13]. Although the transition from egg laying to live-bearing has repeatedly arisen, whether phenotypic evolution in viviparous lineages is also repeatable remains unclear. For example, to what extent are phenotypic shifts correlated or decoupled among transitions to live birth? And, how strongly do phenotypic patterns among different viviparous lineages relate to the thermal environment and to the cold-climate hypothesis?

One key lens with which to approach these questions is by examining the ratio by which energy is acquired and allocated to survival, growth, and reproduction (metabolic rate), and the energy allocated to the number and size of hatchlings (production). Compared to oviparous counterparts, viviparous species often exhibit lower mass-specific metabolic rates[14,15] and reduced mass-specific production[16]. Nonetheless, it remains unclear whether mass-specific metabolic rates and mass-specific production shift in tandem or independently of each other in viviparous lineages. Metabolic rate increases with body mass and body temperature (see equation in Methods)[17,18]. Shifts in morphology and thermal physiology, therefore, can produce several trait combinations that result in a lower mass-specific metabolic rate[18] in viviparous species (Table 1a). In another way, mass-specific production is determined by the trade-off between offspring mass and the number of hatchlings or eggs produced yearly (annual fecundity) divided by female body mass[16,19]. Thus, the lower mass-specific production of viviparous species may reflect different combinations of trait shifts (Table 1b). What combinations between body mass and body temperature are associated with patterns of mass-specific metabolic rate in viviparous linages? Likewise, are shifts in offspring mass and/or offspring number (annual fecundity) evolving in tandem or independently among viviparous lineages? More broadly, how does thermal physiology (beyond just field body temperatures for activity) evolve in viviparous lineages and, in light of the CCH, how strongly do shifts in the thermal environment predict parity mode evolution?

The repeated origin of viviparity among closely-related species provides a naturally replicated framework in which to test for shared signatures of adaptation. Squamate reptiles (lizards and snakes) account for ~75% of the origins of viviparous vertebrates[7]. Here, we focused on phrynosomatid lizards, a lineage well known for repeated transitions to live birth[20,21], to investigate the associations between behavior, physiology, morphology, and life history associated with viviparity. This diverse lizard family is comprised of 170 species distributed from North to Central America, and at elevations ranging from sea level to nearly 5000 m[21–23]. We assembled a dataset of adult body mass, adult body size (snout-vent length; SVL), thermoregulatory behavior and thermal physiology (field-estimated active and inactive body temperatures, laboratory preferred temperatures, field-measured thermoregulatory effectiveness, and critical thermal limits), metabolic physiology (mass-specific, mass-corrected, and temperature-corrected metabolic rate), and life-history traits (offspring mass, offspring size, clutch/litter size, and annual mass-specific production) by combining newly collected with previously published data from 125 phrynosomatid species (80 oviparous and 45 viviparous species). To assess how phenotypic variation relates to the environments species occupy, we also estimated both broadscale (environmental layers at 1 km² resolution) and fine-scale (operative environmental temperatures ($T_e$) during activity and inactivity periods) thermal variation for each lizard population. Our dataset includes 73% of phrynosomatids and representatives from all viviparous sub-lineages. We then fitted a series of evolutionary models to the behavioral, physiological, morphological, and reproductive data to determine the patterns of trait evolution associated with oviparity and viviparity, and assess the strength of phenotypic convergence in viviparous species. Through a series of regression approaches, we then investigated the evolutionary relationships between environmental temperatures and parity mode shifts, and how thermoregulatory behavior varies across parity modes and thermal environments.

Here, we show that the evolution of viviparity in phrynosomatids is associated with convergent reductions in cold tolerance, field body temperature, laboratory preferred body temperature, heat tolerance, mass-specific metabolic rate, annual number of offspring, and mass-specific production. Viviparous species maintain behaviorally lower body temperatures of activity (even in warm habitats) and have a lower fecundity than oviparous species, which reduces their energetic burden allocated to maintenance and reproduction. Together, our results indicate that viviparity represents the slow end of the fast-slow life-history

---

**Table 1 Three trait combinations could explain the lower mass-specific metabolic rate of viviparous species (a) and three other trait combinations could explain their lower mass-specific production (b).**

| | **(a) Trait combinations resulting in a lower mass-specific metabolic rate** |
|---|---|
| i | Body mass is similar among oviparous and viviparous species, but body temperature is lower in viviparous species. |
| ii | Body mass is higher and body temperature is lower in viviparous species. |
| iii | Body mass is higher in viviparous species, but body temperature is similar among oviparous and viviparous species. |

| | **(b) Trait combinations resulting in a lower mass-specific production** |
|---|---|
| iv | Offspring size is similar among oviparous and viviparous species, but annual fecundity is lower in viviparous species. |
| v | Offspring size is smaller and annual fecundity is lower in viviparous species. |
| vi | Offspring size is smaller in viviparous species, but annual fecundity is similar between oviparous and viviparous species. |

Note that phrynosomatids are ancestrally oviparous and there are no back-transitions to oviparity. Therefore, our trait combinations are structured around explaining the transition to viviparity (rather than the other way around).

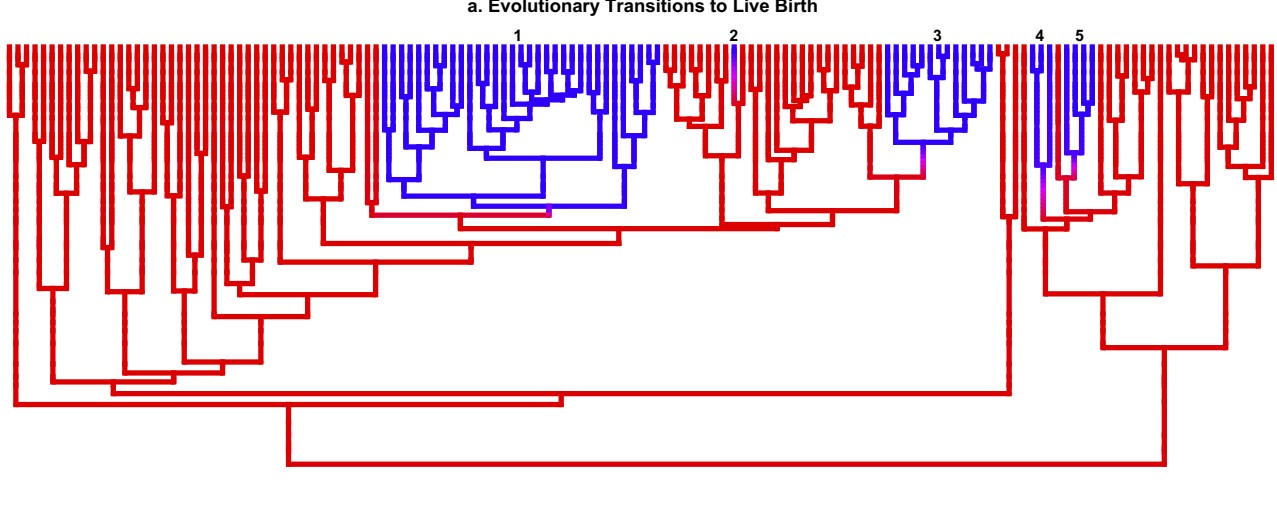

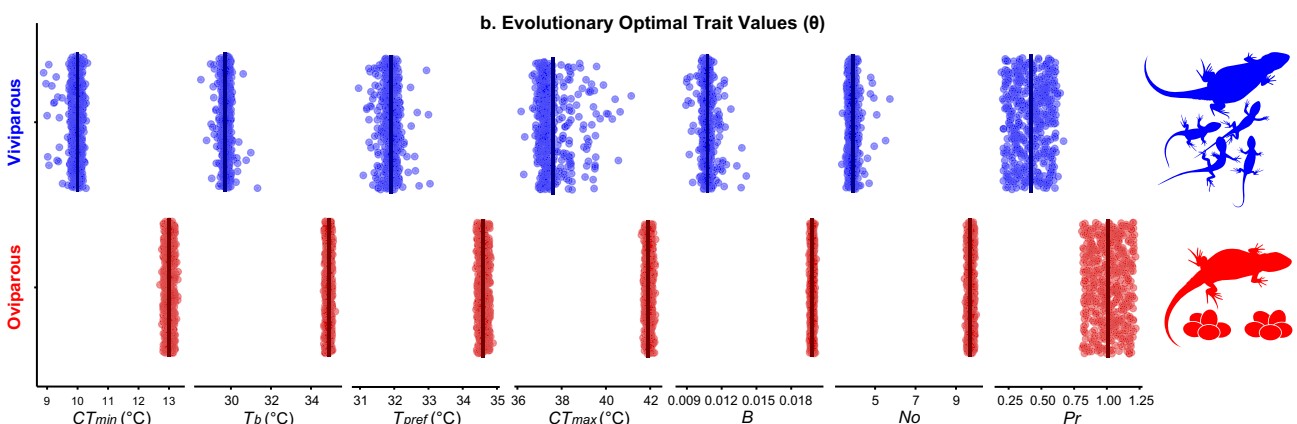

**Fig. 1 Parallel reductions in physiological and life-history traits are associated with viviparity in phrynosomatid lizards. a** Five evolutionary transitions from egg laying (red) to live-bearing (blue) occurred in phrynosomatids. **b** Viviparous lineages are characterized by reductions in the evolutionary optimal trait values (θ) for cold tolerance ($CT_{min}$), field-estimated body temperature ($T_b$), the laboratory-measured preferred body temperature ($T_{pref}$), heat tolerance ($CT_{max}$), mass-specific metabolic rate during activity (B), the annual number of offspring (No), and mass-specific production (Pr). Evolutionary optimal trait values were inferred from the Ornstein-Uhlenbeck (OU) model-fitting procedure (see Methods). Each point represents a different stochastic character map from our analyses across the maximum clade credibility tree (n = 500 per trait). These patterns are matched with a strong signal of phenotypic convergence among viviparous species (Supplementary Table 6). Source data are provided as a Source Data file.

continuum and that behavioral and physiological changes associated with viviparity facilitate access to cold habitats. This study sheds new light on the multidimensional patterns of evolution associated with viviparity and, through their interpretation, the factors that underpin its origin in squamates.

## Results and discussion

We began by building a phylogeny for phrynosomatids (Supplementary Fig. 1) and reconstructing parity mode across the tree. Our reconstructions strongly support five independent shifts to live-bearing, and no back-transitions to oviparity (Fig. 1a; Supplementary Fig. 2). Transitions from egg laying to live birth in phrynosomatids are strongly associated with an 1.8-fold reduction in the evolutionary optimum (θ parameter; see Methods) for mass-specific metabolic rate during activity (Fig. 1b; Supplementary Table 2). Likewise, across multiple transitions from oviparity to viviparity we detected a 2.4-fold reduction in optimal annual production (Supplementary Table 2). Put together, viviparity in phrynosomatids represents a multidimensional phenotype in which the ratio and quantity of energy allocated to

maintenance and production are decreased, a finding robust to repeated origins of live birth (Supplementary Table 6). These evolutionary patterns also align with findings from biophysical approaches, namely that viviparous females have a lower embryonic energy consumption than oviparous females[9].

Our results can be interpreted via a combination of metabolic and life-history theory[15,18,19,24]. In low-predation environments, populations evolve toward a lower metabolic rate and lower reproductive allotment[24]. Species with lower mass-specific metabolic rates also exhibit reduced mass-specific production and are positioned on the slow end of the fast-slow life-history continuum[15]. Given these premises and assuming steady-state populations—populations in which energy invested into production (birth rates) equals energy lost by predation (death rates)[19,25]—viviparity in phrynosomatids is a high-survivorship, low-fecundity phenotype positioned on the slow end of the fast-slow continuum. This 'slow' life-history strategy is characterized by the reduction in mortality afforded by in utero embryonic development (in comparison to eggs deposited in nests) against abiotic and biotic hazards[2–5,8,26–31] and is favored in colder environments such as high elevation[8,20,21], where predation risk

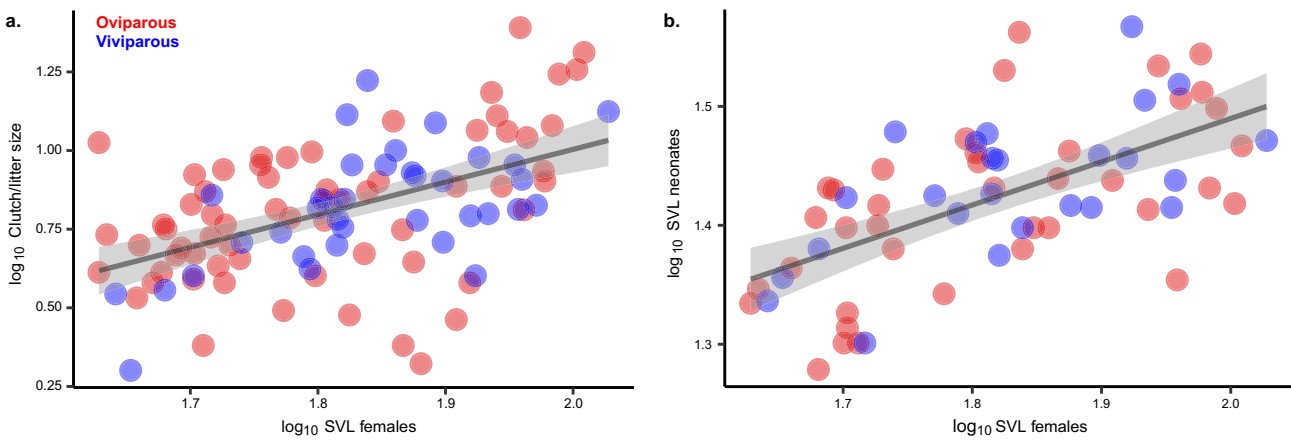

**Fig. 2 Dam's body size is positively correlated with clutch/litter size and neonate body size in phrynosomatid lizards.** Influence of dam's body size on clutch/litter size (**a**; y = 1.1321x − 1.2362, two-sided P < 0.001, n = 64 oviparous species and 36 viviparous species) and on the body size of neonates (**b**; y = 0.2893x + 0.9064, two-sided P < 0.001, n = 39 oviparous species and 25 viviparous species). Data are presented as mean values, and regression lines are wrapped by 95% confidence bands. The regression slopes were estimated by phylogenetic regressions (PGLS), and the source data are provided as a Source Data file.

for phrynosomatids and other ectotherms is lower[32–34]. Furthermore, when compared to their oviparous counterparts, viviparous phrynosomatids are more common in tropical environments with lower temperature seasonality[21], where selection could favor allocating energy in fewer, fitter offspring, rather than higher productivity[35]. Whereas viviparous females can replace themselves each generation by allocating less energy to maintenance and production (normalized by body mass), oviparous females instead expend greater energy on producing more eggs per year (Fig. 1b; Supplementary Table 2).

**Pathways for a reduced mass-specific metabolic rate.** Shared reductions in mass-specific energetic demand could reflect different evolutionary pathways involving changes in body mass, field body temperature, or both (Table 1a). Across five transitions to viviparity, we do not find any support for adaptive shifts in body mass associated with the reproductive mode in phrynosomatids (Supplementary Table 2). Phylogenetically-corrected body size (SVL), which is strongly correlated with age at sexual maturity[12] and with body mass ($\log_{10}$body mass = 0.288$\log_{10}$SVL + 1.522, P < 0.001; Supplementary Table 4), was positively correlated with clutch/litter size ($\log_{10}$clutch or litter size = 1.132$\log_{10}$SVL −1.236, P < 0.001; Fig. 2a; Supplementary Table 4) and offspring size ($\log_{10}$offspring size = 0.289$\log_{10}$SVL + 0.906, P < 0.001; Fig. 2b; Supplementary Table 4). Therefore, whether oviparous or viviparous, larger females are more fecund, and produce larger neonates[36,37]. Correspondingly, we infer that, during the transition to living birth in phrynosomatids, the evolutionary optimal body size in females[38] (θ = 61.1 mm SVL; Supplementary Table 3) likely remains unchanged because any size shift would also affect the quantity and quality of offspring.

In contrast to body mass, however, transitions to viviparity are associated with a 5 °C reduction in the optimal field-active body temperatures (θ = 29.7 °C) when compared to their oviparous counterparts (θ = 34.9 °C; Fig. 1b; Supplementary Table 2). Therefore, reductions in core temperature, but not body mass, characterize metabolic changes in the transition to living birth in phrynosomatid lizards (trait combination i in Table 1a). That finding is further supported by our estimated mass-corrected and temperature-corrected metabolic rate: both parity modes share the same optimal temperature-corrected metabolic rate, but viviparity is associated with an 0.9-fold reduction in the optimal mass-corrected metabolic rate of activity (Supplementary Table 2). This finding, based on field-

active body temperatures, also holds when considering energetic demand during inactivity: the mass-corrected metabolic rate was lower for viviparous species than their oviparous counterparts (Supplementary Table 2). Given that overly high incubation temperatures negatively affect embryos[3,39,40], reductions in the field-active body temperatures of viviparous species could optimize intrauterine embryonic development[41]. For example, when pregnant *Sceloporus jarrovii* females are maintained at 36 °C, neonates are smaller and offspring mortality reaches ~50%[42]. By contrast, in female *S. jarrovii* that maintain a body temperature of ~32 °C neonates are larger and offspring viability is >95%[42]. Indeed, during pregnancy (especially late pregnancy) in phrynosomatids body mass and metabolic rate of females increase, and females tend to behaviorally compensate by lowering their body temperature[41,43–48]. For example, a behavioral reduction of 2.5 °C in the preferred temperature of pregnant *S. jarrovii* likely alters their metabolic rate to that of non-pregnant lizards[41].

**Lower fecundity reduces mass-specific production in viviparous lizards.** In addition to energetic adjustments mediated by thermal behavior, we detected convergent shifts to lower mass-specific production in viviparous species (Supplementary Table 6). Reductions in mass-specific production might reflect different combinations of offspring mass and annual fecundity (Table 1b). Yet, the evolutionary optima for offspring mass (θ = 0.83 g; Supplementary Table 2) and offspring size (θ = 26.4 mm snout-vent-length; Supplementary Table 2) do not vary between viviparous and oviparous species. The retention of optimal offspring size[12] and offspring mass in viviparous lineages could reflect the presence of a shared adaptive optimum: indeed, empirical studies on phrynosomatids have found that excessively small or large offspring (based on maternal morphology) typically exhibit reduced survivorship[49,50].

As described above, the offspring body size is highly correlated with dam's body size (Fig. 2b), but, crucially, the temperature during embryogenesis also impacts neonate body size[39,40,42]. Although viviparous species certainly exhibit lower body temperatures when compared to their oviparous relatives, those body temperatures are nonetheless, on average, considerably higher (mean $T_b$ = 31.2 °C, n = 38) than those available in their environments (mean $T_e$ = 26.0 °C, n = 28; U = 293.5, two-sided P = 0.002). This finding is consistent with the 'cold-climate hypothesis': females can thermoregulate above ambient

temperatures in cold environments to shorten embryonic development time and reduce exposure to overly cold temperatures[3,9]. Compared with embryos developed in the dam's uterus, eggs in nests in cold habitats are exposed to lower incubation temperatures[9]. Depending on the temperature, eggs could be viable at low temperatures, but offspring may be smaller, likely related to a longer incubation and consequently high energy consumption[9,39,40]. Therefore, in cold habitats, viviparity helps maintain the optimal offspring size, whereas oviparity could induce smaller (i.e., poorer quality) neonates[39]. This effect of intrauterine incubation could represent one of the main reproductive advantages of live-bearing species in cold habitats (consistent with the CCH)[3].

Nonetheless, when compared with viviparous species, oviparous phrynosomatids produce ~2.5-fold more offspring per year (Fig. 1b; Supplementary Table 2). The lower annual fecundity and consequent lower mass-specific production in viviparous species could reflect their relatively long gestation periods, which limit most species to a single litter per year[16,51]. Therefore, the available evidence indicates that in the evolutionary transition to viviparity, selection favors allocating less energy into production, while leaving offspring size unchanged (corresponding to trait combination number iv in Table 1b). As in many other lineages, the differences we observe here among oviparous and viviparous lineages highlight the evolutionary tightrope organisms face between the competing fitness demands of producing high-quality offspring (i.e., high survivorship) and producing more offspring (i.e., high fecundity). For example, a strong association between viviparity and lower fecundity is not only common in squamates but also occurs in gastropods, insects, and fishes, in which lower fecundity is counterbalanced by higher offspring quality and survivorship[1,2,52,53].

**Patterns and mechanisms of physiological evolution in viviparous lizards.** Although viviparous species are conspicuously more prevalent in cooler habitats than oviparous species, they certainly also occur in warm environments[20,21]. Whether found in relatively cool habitats or in relatively warm habitats (Supplementary Fig. 3), viviparous lizards generally exhibit a lower core temperature than their oviparous counterparts (see Supplementary Data 1). Correspondingly, the field-measured body temperature of viviparous species is poorly correlated with mean annual temperature (Fig. 3b), or with any of our environmental variables (Supplementary Table 4). By contrast, mean annual temperature is positively correlated with body temperature in oviparous lizards (Fig. 3b), although the strength of the relationship is weak (Supplementary Table 5).

Regardless of parity mode, thermoregulatory effectiveness (i.e., the ability to maintain the core temperature within the preferred range) is uniformly high, and statistically indistinguishable among parity modes in phrynosomatid lizards ($U = 467.5$, two-sided $P = 0.67$, $n = 37$ oviparous species and 27 viviparous species; mean $E = 0.8 \pm 0.02 SE$ for both modes). Even when viviparous species are found in warm habitats and oviparous species are found in cool habitats, phrynosomatid lizards are effective at maintaining their field body temperature within (or close to) their respective preferred thermal ranges. For example, *Sceloporus bulleri, S. macdougalli, S. prezygus, S. serrifer,* and *S. stejnegeri* are viviparous lizards inhabiting relatively warm environments: these species nonetheless maintain a lower preferred body temperature (see Supplementary Data 1). By contrast, *Sceloporus aeneus, S. graciosus, S. slevini,* and *S. vandenburgianus* are oviparous lizards from relatively cold environments, and they exhibit a higher preferred body

temperature than viviparous counterparts in similar habitats (see Supplementary Data 1).

Furthermore, under the threshold model, we found weak evolutionary covariation between environmental predictors and reproductive mode (mean annual temperature: $r = -0.205$; the mean temperature of the coldest quarter: $r = -0.001$; the mean temperature of the warmest quarter: $r = -0.359$), implying that environmental temperature is not strongly associated with evolutionary transitions to viviparity. Likewise, phylogenetic logistic regressions[54] indicate that viviparity is not predicted by mean annual temperature ($z = -0.367$, $P = 0.7$, $n = 56$ oviparous species and 38 viviparous species), mean temperature of the coldest quarter ($z = -0.0056$, $P = 1$, $n = 56$ oviparous species and 38 viviparous species), or mean temperature of the warmer quarter ($z = -0.849$, $P = 0.4$, $n = 56$ oviparous species and 38 viviparous species). Given these results, we infer that cool-adjusted thermal physiology of viviparous species can facilitate access into cooler environments, but that evolution of viviparity need not be a necessary outcome of shifts into cooler environments.

While evolutionary covariation between the thermal environment and reproductive mode is weak, the degree of evolutionary convergence towards a more cool-adjusted thermal physiology in viviparous lizards is remarkably strong (as indicated by the Wheatsheaf index; Supplementary Table 6). The evolutionary optimum for cold tolerance is 3 °C lower in viviparous phrynosomatids ($\theta = 10.0$ °C) than in their oviparous counterparts ($\theta = 13.0$ °C; Fig. 1b; Supplementary Table 2). Likewise, the phenotypic optimum for heat tolerance is 4.3 °C lower in viviparous ($\theta = 37.6$ °C) than oviparous ($\theta = 41.9$ °C) species (Fig. 1b; Supplementary Table 2). In addition to thermal limits, the evolutionary optimum for the preferred body temperature ($T_{pref}$) is lower in viviparous species than their oviparous relatives ($\theta = 31.9$ °C for viviparous and 34.6 °C for oviparous species; Fig. 1b; Supplementary Table 2), although the degree of convergence for $T_{pref}$ was weaker (Supplementary Table 6).

Among these shifts to more cool-adapted physiology in viviparous species, only cold tolerance reflects adjustments to cooler environments. In particular, we found a strong positive relationship between mean annual temperature and cold tolerance in both oviparous and viviparous lineages (Fig. 3a; Supplementary Table 4). This relationship is matched by an instantaneous pace of cold tolerance adaptation ($t_{1/2} = 0$ million years, and $\alpha = \infty$ for both viviparous and oviparous species; Supplementary Table 5). Thus, the reduced cold tolerance of viviparous species likely arises from the fact they are more prevalent in cooler environments than oviparous phyrnosomatids[20,21]. Enhanced cold tolerance in cooler environments, regardless of parity mode, fits into a broader picture of ecophysiological evolution in ectotherms: cold tolerance is phylogenetically labile[55] and often rapidly adapts to the minimum temperatures ectotherms experience in their environments[56]. By contrast, heat tolerance adapts slowly to the thermal environment itself (Fig. 3d; Supplementary Table 4), in turn reflecting a much longer phylogenetic half-life for this trait ($t_{1/2} = 8.8$ million years, and $\alpha = 0.08$ for viviparous, and $t_{1/2} = 17.8$ million years, and $\alpha = 0.04$ for oviparous species; Supplementary Table 5). Likewise, mean annual temperature is a weak predictor of preferred body temperature both for viviparous and oviparous species (Fig. 3c; Supplementary Table 4), reflecting protracted lags in adaptation ($t_{1/2} = 13.8$ million years, and $\alpha = 0.05$ for viviparous, and $t_{1/2} = 9.7$ million years, and $\alpha = 0.07$ for oviparous species; Supplementary Table 5).

Taken together, our results imply that the thermal behavior and physiological properties of viviparous species are not exclusively by-products of live-bearing species being more common in colder environments. Instead, these patterns are

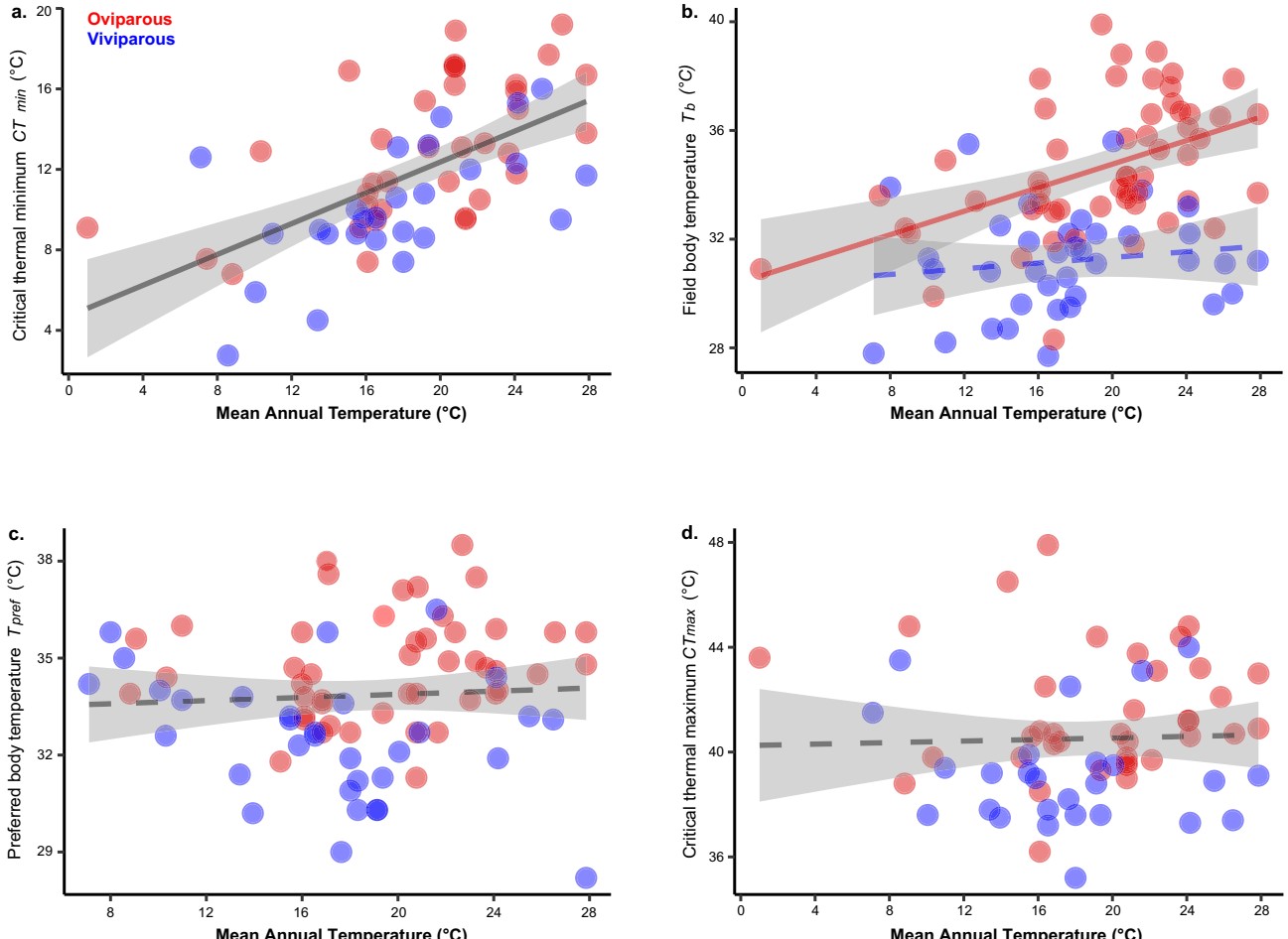

**Fig. 3 Phylogenetic regressions between the thermal environment, and the thermoregulatory behavior and thermal physiology in phrynosomatid lizards.** Influence of mean annual temperature (MAT) on cold tolerance (**a**; $y = 0.354x + 5.639$, two-sided $P < 0.001$, $n = 36$ oviparous species and 26 viviparous species), field body temperature (**b**; for oviparous: $y = 0.230x + 30.3$, two-sided $P < 0.001$, $n = 55$, and for viviparous: $y = 0.055x + 30.69$, two-sided $P = 0.4$, $n = 37$), preferred body temperature (**c**; $y = -0.0202x + 35.05$, two-sided $P = 0.6$, $n = 47$ oviparous species and 32 viviparous species), and heat tolerance (**d**; $y = 0.032x + 41.29$, two-sided $P = 0.5$, $n = 37$ oviparous species and 26 viviparous species). Solid lines represent slopes with statistical significance (<0.05), and dashed lines represent slopes that are not statistically different from 0. Data are presented as mean values, and 95% confidence bands are included around regression lines. The regression slopes were estimated by phylogenetic regressions (PGLS), and the PGLS results using the other macroclimatic predictors (bio10 and bio11) and operative temperatures ($T_e$) are given in Supplementary Table 4. Parity mode did not impact the relationship between MAT and $CT_{min}$ (**a**), $T_{pref}$ (**c**), or $CT_{max}$ (**d**); therefore, data were combined for oviparous and viviparous species (as indicated by a single gray regression line). By contrast, parity mode impacts the relationship between MAT and $T_b$ (**b**), as the $P$-value (0.02) was below the Bonferroni-corrected significance level of 0.025. Source data are provided as a Source Data file.

also part and parcel of a broader set of life-history adjustments altering how energy is allocated to growth, maintenance, and reproduction. Given that viviparity and oviparity are on different sides of the 'slow-fast' life-history continuum, transitions to live birth may reflect a number of potential selective trade-offs important for life-history evolution, including (but not limited to) environmental temperature, hypoxia at high elevation, lower food availability, higher intra- and interspecific competition, and high predatory risk in eggs[1,2,5,57]. In this sense, the 'cold-climate hypothesis' fits into a broader set of energetic trade-offs and selective pressures that may favor the evolution of viviparity. While the inference space of our results is limited to phrynosomatid lizards, the general principles that explain evolutionary patterns in this system also characterize other ectotherm lineages, which account for nearly all origins of viviparity in animals. We suspect, therefore, that the patterns we observed here might be generalizable across the animal tree of life.

## Methods

**Ethics statement.** The data collection and experiments were conducted in accordance with the collecting permits (SGPA/DGVS/07946/08, 03369/12, 00228/13, 07587/13, 01629/16, 01205/17, 02490/17, 06768/17, 000998/18, 002463/18, 002490/18, 002491/18, 003209/18, and 02523/19) approved by Dirección General de Vida Silvestre, México.

**Phylogeny and divergence time estimation.** To estimate the phylogeny and divergence time among phrynosomatid species we used sequences of five mitochondrial and eight nuclear genes available in GenBank for 149 taxa (Supplementary Data 2). Accession numbers were the same as those used in Martínez-Méndez et al.[58] for the *Sceloporus torquatus, S. poinsettii* and *S. megalepidurus* groups and the same as those in Wiens et al.[59] for other phrynosomatid species. For taxa not included in the previous references, we searched GenBank for available sequences. We then performed alignments for each gene using MAFFT (ver. 7)[60] and concatenation and manual refinement using Mesquite (ver. 3.6);[61] obtaining a concatenated matrix of 9837 bp for 149 taxa (Supplementary Data 3). For the relaxed clock analyses, three nodes were calibrated using lognormal distributions based on two previous studies[59,62]. The first calibration was set for the *Sceloporus* clade (offset 15.97 million years ago (MYA)) based on a fossil *Sceloporus* specimen[63]). The second calibration point was set for the *Phrynosoma* clade (offset 33.3 MYA) based on the fossil *Paraphrynosoma greeni*[64], and the last calibration

point was for the *Holbrookia-Cophosaurus* stem group (offset 15.97 MYA) given the fossil *Holbrookia antiqua*[63]. We conducted dating analysis with the concatenated sequences matrix, partitioned the mitochondrial and nuclear information, each gene under GTR + I + Γ model, and allowed independent parameter estimation. We performed Bayesian age estimation with the uncorrelated lognormal relaxed clock (UCLN) model in BEAST (ver. 2.5.2)[65,66] and run on CIPRES[67]. Tree prior (evolutionary model) was under the Birth-Death model, and we ran two MCMC analyses for 100 million generations each and stored every 20,000 generations. We assessed the convergence and stationarity of chains from the posterior distribution using Tracer (ver. 1.7)[68]. We combined independent runs using LogCombiner (ver. 2.5.2; BEAST distribution)[69] and discarded 30% of samples as burn-in, obtaining values of effective sample size (ESS) greater than 200. We estimated the maximum clade credibility tree from all post-burnin trees using TreeAnnotator (ver. 1.8.4)[69]. The ultrametric tree is available as Supplementary Data 4. As we describe below, we accounted for phylogenetic uncertainty in our models by reperforming analyses using 500 trees that we randomly sampled from our posterior distribution. The 500 sampled trees are available as Supplementary Data 5.

## Data collection

*Parity mode.* We categorized each species as either oviparous or viviparous based on previously published databases[21,37,51,70], published references, and unpublished data (Supplementary Data 1). Our assignments align with other studies, except for one species, *Sceloporus goldmani*, which has been previously considered a viviparous species[21,71–73]. The only available sequence in GenBank (U88290) for that species is from a male (MZFC-05458) collected in Coahuila, Mexico[72]. However, in that same locality, one of us (F. R. Méndez-de la Cruz; unpubl. data) collected two females of the same species, and both laid eggs. Thus, the population of *S. goldmani* herein included is considered oviparous. Considering *S. goldmani* viviparous increases the number of originations of viviparity to 6 (from 5) in this lineage (Supplementary Fig. 4), but does not alter the outcome of our model-fitting analyses of trait evolution (Supplementary Table 7).

*Thermal physiology.* We compiled a database of four thermal physiological traits that influence the performance and fitness of ectotherms[74] for 104 phrynosomatid species. These data were gathered from both published sources and from our own field and laboratory work (Supplementary Data 1). The thermal physiological traits we examined were the field body temperature ($T_b$) of active lizards, the preferred body temperature ($T_{pref}$) in a laboratory thermal gradient[75], cold tolerance (critical thermal minimum, $CT_{min}$), and heat tolerance (critical thermal maximum, $CT_{max}$). These latter two traits ($CT_{min}$ and $CT_{max}$) describe the thermal limits of locomotion; specifically, they describe the lower and upper temperatures, respectively, at which lizards fail to right themselves when flipped onto their backs[55,76]. To minimize the confounding effects of experimental design, we limited our data selection to species that were measured with similar methods. Correspondingly, our new data collection approach mirrored that of the published studies from which we extracted data. To obtain mean values for each thermal physiological trait ($CT_{min}$, $T_b$, $T_{pref}$ and $CT_{max}$) we did not mix data measured from different locations (instead, we used data from the population with the highest sample size).

For species that we newly measured thermal physiological traits, we obtained the data as we describe below, and we based our methodology on the previous work[55,56,75,76]. We captured active (perching) adult lizards by lasso or by hand, and immediately (<10 s) we measured their body (cloacal) temperature using a thermocouple (type K) connected to a digital quick-reading thermometer. We transported lizards to a field laboratory (which was at an ambient temperature of ~20 °C), measured the SVL and body mass of individuals, and the next day we measured their preferred body temperature by placing them into a laboratory thermal gradient from 8:00 to 17:00 h. The laboratory thermal gradient consisted of a wooden box (100 cm wide, 100 cm long, and 30 cm tall) divided into ten tracks. At one extreme of the laboratory thermal gradient, we placed ten 75 W bulbs (one per track) at the height of 25 cm above to generate a thermal gradient ranging from ~50 °C in the hot extreme to ~20 °C in the cold extreme. Then, we placed each lizard on a track, and we measured their cloacal temperature (using the same thermocouple and thermometer as in the fieldwork) every hour, during the length of the experiment. When we finished the experiment on thermal preferences, we performed the experiment on heat tolerance. For that, we placed individually each lizard into a plastic container (25 cm diameter and 30 cm height) and we increased their body temperature (1 °C/m) using a 90 W bulb suspended ~40 cm above the container. When lizards initiated panting behavior, we began flipping them onto their back every 20 s, and we recorded $CT_{max}$ as the body temperature at which a lizard lost the ability to right itself. On the next day, we each placed individually each lizard into a plastic container (23 cm wide, 16 cm long, and 8 cm tall), and the plastic container on a bed of ice. Every 20 s we began flipping lizards onto their back, and we recorded $CT_{min}$ as the body temperature at which a lizard lost the ability to right itself. For $CT_{max}$ and $CT_{min}$ experiments, we did not include pregnant/gravid females, and after laboratory experiments, we hydrated lizards ad libitum, and released them at their capture sites.

In Supplementary data 1 we indicate coordinates where thermal physiological traits were measured and the environmental variables associated with each coordinate. In cases where locality details, but not coordinates, were available, we

georeferenced sampling sites using Google Earth Pro (Version 7.3.3). All physiological data correspond only to adult lizards. Some studies have found that pregnant females reduce their core temperature to better match the optimal incubation temperature for their offspring[41,43,48]. When we detected effects of reproductive condition on $T_b$ or $T_{pref}$, we excluded data from pregnant (or gravid) females. To test whether behavioral and physiological traits differed between sexes, we performed t-tests for a sub-set of 25 species (Supplementary Table 1). We did not find significant behavioral and thermal physiological differences between (non-gravid/non-pregnant) females and males in $T_b$ ($t = 0.172$, $df = 48$, $P = 0.86$), $T_{pref}$ ($t = −0.482$, $df = 48$, $P = 0.63$), $CT_{min}$ ($t = 0.742$, $df = 45$, $P = 0.46$), or $CT_{max}$ ($t = −0.407$, $df = 42$, $P = 0.69$), so we combined data for both sexes. Ideally, we would rerun all analyses using thermal trait data from gravid/pregnant females, but such data are still lacking. Given that, in the few cases where robust data do exist, preferred temperatures in pregnant females tend to be even lower than in non-pregnant females[46,47,77], we suspect that our analyses provide a relatively conservative estimate of physiological differences among parity modes.

*Operative temperatures.* As we describe below, we were interested in estimating thermoregulatory patterns among phrynosomatid species. Doing so requires knowledge of the environmental operative temperatures ($T_e$) available to lizards. $T_e$ represents the equilibrium temperature of an animal in the absence of behavioral thermoregulation[78]. We recorded $T_e$ using previously-calibrated pipe models (made of polyvinylchloride), which were similar in shape, size, and heat gain/loss with respect to lizards of each species (for examples of calibration, see refs. [56,78,79]). Into each pipe model, we inserted one temperature data logger (Thermochron iButton; model DS1921G), which recorded temperature (±0.1 °C) every ten minutes during the same periods (and days) during which we were measuring field-active body temperatures ($T_b$) in lizards. These models were placed randomly in microsites occupied by lizards in their activity period[56,79]. Operative temperatures were typically measured during a sampling period of 1–5 days for each locality, which always occurred during times of the year when lizards exhibit surface activity. The pipe models also recorded temperature during the night ($T_{e-night}$), which we used as an approximation of core temperatures of individuals during their inactivity period to model mass-corrected metabolic rates during inactivity (described below). As a caveat, the $T_{e-night}$ measured in microsites where individuals were active are likely to be somewhat cooler than microenvironmental temperatures experienced by inactive lizards in their retreats.

*Thermoregulatory effectiveness.* Several studies have found that viviparous species have lower field body temperatures than their oviparous counterparts[11,80]. Less well known, however, is whether lower field body temperatures reflect a behaviorally passive property of viviparous lizards, perhaps because of their distributions in relatively cooler habitats, or whether those low field body temperatures reflect a more behaviorally active decision to thermoregulate. Therefore, we were particularly interested in the thermoregulatory patterns of oviparous and viviparous species. We calculated the effectiveness of temperature regulation ($E$), a ratio that describes how well lizards maintain their $T_b$ within their $T_{set}$ range (central 50% of data of $T_{pref}$, $T_{set25}$, and $T_{set75}$), given the operative temperatures ($T_e$) available in their habitat[75]. We estimated $E$ for each species following the equation proposed by Hertz et al.[75]:

$$E = 1 - (\overline{db}/\overline{de}) \qquad (1)$$

where $\overline{db}$ is the average of the accuracy of body temperature, and indicates the deviation of $T_b$ from $T_{set}$ range. If each $T_b < T_{set25}$, then each $db = T_{set25} - T_b$, if each $T_b > T_{set75}$, then each $db = T_b - T_{set75}$, and if each $T_b$ is within $T_{set}$ range, then each $db = 0$. $\overline{de}$ is the average of thermal quality of the habitat, and indicates the deviation of $T_e$ from $T_{set}$ range. If each $T_e < T_{set25}$, then each $de = T_{set25} - T_e$, if each $T_e > T_{set75}$, then each $de = T_e - T_{set75}$, and if each $T_e$ is within $T_{set}$ range, then each $de = 0$. Values of $\overline{db}$ close to 0 indicate that lizards accurately maintain their body temperature within their preferred range, and values of $\overline{de}$ close to 0 indicate that the habitat temperatures approximate (and/or fall within) the preferred range of lizards. As both $\overline{db}$ and $\overline{de}$ increase, body temperatures and operative temperatures, respectively, exceed species' preferred thermal ranges. As such, $E$ values close to 1 indicate that lizards are highly effective thermoregulators, and $E$ values close to 0 indicate that individuals are more behaviorally passive with respect to the thermal environment. $E$ was only estimated in cases where $T_e$ and $T_b$ were sampled during the same period, and if $T_{set}$ was measured from the same population of lizards from which $T_b$ was measured. In total, we were able to gather estimates of $E$ from 64 species (37 oviparous and 27 viviparous) of phrynosomatid lizards (Supplementary Data 1).

*Environmental temperature.* In addition to the operative temperatures, which provided a detailed (but temporally limited) snapshot of the thermal environment, we gathered data on general air temperature trends for each species' habitat. Specifically, we also gathered climatic measurements for each locality (Supplementary Data 1) from which any lizard trait data were gathered by extracting thermal variables from the environmental layers available in the WorldClim dataset (resolved to ~1 km²)[81]. These variables were mean annual temperature (bio1), mean temperature of the warmest quarter (bio10), and mean temperature of the

coldest quarter (bio11). We did not use these data to calculate $T_e$ for estimates of thermoregulatory effectiveness (as $E$ should be calculated from $T_e$ measured during the same time period as $T_b$). Instead, we used these bioclimatic variables as predictors of phenotypic trait variation using evolutionary regressions as described below.

*Morphology and life-history traits*. We gathered published and unpublished information for mean snout-vent length (SVL; mm), a common measure of body size in squamates, and body mass (g) of adult females and neonates. We also recorded clutch or litter size (i.e., the number of offspring produced per reproductive bout), and the number of clutches or litters produced during 1 year (Supplementary Data 1). We multiplied these two last traits to quantify annual fecundity, which reflects the total predicted annual reproductive output of a given species. We used annual fecundity for three reasons. First, in phrynosomatids (with the exception of some populations of three species[82–84]), females have annual (seasonal) patterns of reproduction[51]. Second, oviparous species tend to produce eggs in multiple clutches per year[85], whereas viviparous species are typically able to produce only one litter in the same unit time[51]. Indeed, viviparous species tend to produce only one litter per year regardless of reproductive window length. For example, both *Phrynosoma hernandesi*, a species with shorter gestation (3 months)[86] and *Sceloporus bicanthalis*, a species with continuous reproduction[83,87], produce a single litter per year. Third, the maximum lifespan for phrynosomatid lizards varies considerably but does not differ between parity modes[12]. For some species, the maximum lifespan in natural conditions is ~1 year (documented for the oviparous species, *Sceloporus aeneus*, and the viviparous species, *Sceloporus bicanthalis*[83,88]), whereas for other species the maximal lifespan can approach ~10 years (documented for the oviparous species *Phrynosoma asio*[89] and for the viviparous species *Sceloporus macdougalli*[90]). Thus, consistent with other studies[15,70], we consider that by standardizing production to one year, we have an estimate of reproductive output that can be readily compared among parity modes.

*Metabolic rate*. We modeled individual metabolic rate ($I$) for female lizards following the equation proposed by ref. [18]:

$$I = i_0 M^{3/4} e^{-E/kT} \quad (2)$$

where $i_0 =$ is a normalization constant, $M$ is the mean body mass (g) of females, $e =$ Euler's number, $E =$ activation energy, $k =$ Boltzmann's constant, and $T =$ mean-field body temperature (in Kelvin)[18]. Previous work has shown that the slope and intercept of the body size ~ metabolic rate relationship vary among vertebrate lineages, but vary much less within lineages (with the conspicuous exception of some amphibian lineages like salamanders)[91]. We used $i_0 = \ln(20.3)$ (normalization constant for reptiles) and an $E$ value = 0.63 (activation energy excluding endotherms in hibernation and torpor)[18]. Then, mass-specific metabolic rate ($B$) can be modeled simply as $I/M$, or following the equation proposed by Gillooly et al.[18,92]:

$$B = i_0 M^{-1/4} e^{-E/kT} \quad (3)$$

The universal validity of this equation has been (rightfully) debated[93–98], but is nonetheless a useful (and very widely applied) approximation of instantaneous energetic demand across a wide variety of physiological studies[17,99–102]. Studies that provide empirical estimates of mass-specific metabolic rate typically do so by keeping the experimental temperature constant. This approach allows researchers to compare the mass-specific metabolic rate among different individuals, populations, and species across shared temperature regimes. Yet, this approach would tell us little about the energetic demands of organisms as expressed in their environments. Here, our goal was to understand how observed body sizes and field activity body temperatures impact the energetic demands of lizards based on their reproductive mode. This approach allowed us to predict how mass-specific metabolic rate should vary based on the observed morphology and thermal physiology of the species (during both activity and inactivity periods; see below), as opposed to comparing the metabolic demands across shared thermal conditions.

Mean body length (SVL) of phrynosomatids lizards is more frequently reported than mean body mass. We built a database of mean body mass and mean SVL of adult females for 30 phrynosomatid species (none were gravid or pregnant) via a combination of unpublished and published information (Supplementary Data 1). Using these data, we built a non-phylogenetic equation to predict $\log_{10}$ mean body mass from $\log_{10}$ mean SVL. Our equation is $\log_{10}$ mean body mass $= 3.355\log_{10}$ mean SVL $-5.065$ ($R^2 = 0.88$, $P < 0.001$). Then, we transformed the $\log_{10}$ mean body mass value into an integer value (mean body mass $= 10^{\log10\text{mean body mass}}$). With our equation, we predicted the mean body mass of females for species for which mean SVL and mean-field body temperature were available. Based on this approach, we obtained a total database of the mass-specific metabolic rates of females for 95 phrynosomatid species (Supplementary Data 1).

Certainly, using mean body mass and mean-field body temperature of each species could under- or overestimate mass-specific metabolic rate ("the fallacy of averages")[101,103]. Likewise, it is also relevant to know the instantaneous energetic metabolic demands of species given the body mass or the field body temperature of individuals. Therefore, we also modeled temperature-corrected metabolic rate for each body mass measured in each adult female from 38 phrynosomatid species, and we modeled mass-corrected metabolic rate for each $T_b$ measured in each individual of 65 species, following the equations proposed by ref. [18]: (4)

Temperature-corrected metabolic rate $= 0.71*\ln$ (mass) $+ 18.02$, and (5) mass-corrected metabolic rate $= -0.69*$temperature $(1/kT) + 20.3$. Because $T_b$ can change throughout the day, this approach includes a diel variation on energetic demand within our analyses. Phrynosomatids are diurnal lizards, and metabolic rates from field-estimated body temperatures reflect energetic demand during lizards' period of activity. To test hypotheses about the energetic demands related to parity mode, it is also relevant to know metabolic rates during times of inactivity (like at night). We did not measure the field body temperature of inactive lizards. However, diurnal lizards exhibit a limited ability to thermoregulate at night (a time when thermal environments also tend to homogenize)[104], and their body temperatures, correspondingly, tend to correlate with equilibrium (operative) temperatures (i.e., $T_{e-night}$)[55,56].

So, we also modeled inactivity mass-corrected metabolic rate for each $T_{e-night}$ recorded for each null model during the inactivity time of each species (typically from 18:00 to 08:00 h) in the same localities where field-active body temperatures were measured for 42 species, following the same equation mentioned above. As noted above, the surface perches from which we gathered operative temperatures are likely a bit cooler than nighttime retreats utilized by phrynosomatids and, correspondingly, should slightly underestimate energetic demand during inactivity. Lastly, we estimated the mean temperature-corrected MR, and mean mass-corrected MR during activity and inactivity for each species (Supplementary Data 1). Thus, our approach considers how mass-corrected metabolic rate varies among parity modes during both active and inactive periods.

*Mass-specific production*. We estimated mass-specific production ($Pr$) as the product of neonate mass and annual fecundity/female body mass[16]. Therefore, $Pr$ describes the amount of energy converted into reproductive effort per year, normalized by maternal body mass.

**Evolutionary analyses**. All evolutionary analyses were conducted using the R environment ver. 3.6.0[105], with the exception of the phylogenetic logistic regressions, which were performed using ver. 4.1.1.

*Stochastic character mapping of parity mode*. To estimate the number of transitions between parity modes we performed stochastic character mapping[106] onto the ultrametric tree of Phrynosomatidae using the *make.simmap* function with 500 simulations and a transition model of equal rates (ER) in phytools (ver. 0.6.99) R package[107]. We selected the ER model of character evolution because was it the least-complex, best-supported model ($\Delta AICC = 1.3$, weight $= 0.26$) in comparison to a symmetrical model (SYM; $\Delta AICC = 1.3$, weight $= 0.26$) and with an all-rates-different model (ARD; $\Delta AICC = 0$, weight $= 0.48$).

*Ancestral state reconstruction*. To fit the mean annual temperature through the Phrynosomatidae tree and graphically show the thermal environment where each population of each species used in this study inhabits, we performed ancestral state reconstruction using *contMap* function in phytools (ver. 0.6.99) R package[107].

*Phylogenetic analyses of variance (ANOVA)*. To test for differences in the effectiveness of temperature regulation, among parity modes we performed phylogenetic ANOVAs using the *phylANOVA* function with 500 simulations in phytools (ver. 0.6.99) R package[107].

*Comparing trait evolution between viviparous and oviparous species*. We were interested in whether transitions to viviparity are associated with predictable phenotypic shifts. To this end, we tested if parity mode ("oviparous" or "viviparous") was associated with different evolutionary patterns of mass-specific metabolic rate, mass-corrected metabolic rate (both during periods of activity and inactivity), temperature-corrected metabolic rate, mass-specific production, body mass and size, thermal physiological traits, and life-history traits by fitting Brownian motion (BM) and Ornstein-Uhlenbeck (OU) models. To do so, we used the R package OUwie (ver. 1.57)[108] and the 500 stochastic character maps of parity mode built with the *make.simmap* function in the R package phytools (ver. 0.6.99)[107]. We fitted three different models. The simplest (BM1) is a single-rate BM model in which a single rate of stochastic trait evolution ($\sigma^2$) was estimated for all Phrynosomatidae, and phenotypic differences among species are proportionate to branch length (or time). The other two models were all adaptive OU models that varied in whether the estimated phenotypic optimum ($\theta$) was either constrained to be equal among parity modes (OU1) or allowed to vary between oviparous and viviparous species (OUM). We fitted these three models separately for each physiological trait ($T_b$, $T_{pref}$, $CT_{min}$, $CT_{max}$, mass-specific metabolic rate, mass-corrected metabolic rate, and temperature-corrected metabolic rate), each morphological variable (adult body mass, adult body size), and each life-history trait (offspring mass, offspring size, annual fecundity, and mass-specific production) (Supplementary Table 2). For these (and all) analyses, body mass, body size, offspring mass, offspring size, and annual fecundity were $\log_{10}$ transformed. We assessed model fit using a modified Akaike information criterion ($AIC_C$) that incorporates a correction for a small sample size[109]. Our approach, which was based on 500 stochastic character maps derived from the MCC tree, allowed us to account for uncertainty in reconstruction across the preferred tree, but could not

account for uncertainty in the phylogeny itself. Therefore, we also repeated our stochastic character mapping across 500 individually sampled trees from the posterior distribution to account for this additional source of phylogenetic uncertainty and repeated all of our OUwie analyses using these 500 sampled trees. Our results in this latter approach are comparable to those using the MCC tree (Supplementary Table 3). Therefore, we present our results from the analyses based on the MCC tree in the main document.

More complex models, like OU models, can be incorrectly favored over simpler models if the statistical power of the analysis is weak[110,111]. To assess the adequacy of our phylogenetic data for model-fitting procedures, we performed simulations to assess the probability of type-I error in model fit. We simulated trait evolution in two ways to be consistent with our OU analyses. First, we first simulated trait evolution 500 times on the MCC tree, and then simulated trait data once for each of the 500 trees from the posterior distribution. We then fitted BM and OU models to the simulated data to determine what percentage of analyses would incorrectly favor OU over BM. False positives were not an issue (Supplementary Table 2 and 3), supporting that we could reasonably fit OU models to our data.

*Testing the strength of convergent evolution.* Given that viviparity repeatedly evolved in phrynosomatids (Fig. 1a), we were interested in estimating the strength of convergence in morphology, thermal and metabolic physiology, and life-history traits of viviparous species. To this end, we estimated the Wheatsheaf index ($w$), which quantifies the strength of convergence (or lack thereof) for continuous traits in focal species[112]. We assigned viviparous species as focal groups (recognized a priori with the ancestral state reconstruction), and we estimated $w$ using the *test.windex* function and 500 bootstrap replicates (to estimate a *P*-value) in the R package windex (ver. 2.0.2)[113]. A high $w$ value indicates stronger convergence, and a *P*-value <0.05 represents that convergence is significantly stronger after accounting for relatedness among species[112,113].

*Phylogenetic Generalized Least Squares (PGLS).* We performed PGLS regressions using the *gls* function in the R package nlme (ver. 3.1.139)[114] to know the evolutionary relationships between adult body size and adult body mass, between reproductive response traits (clutch size (or litter size) and offspring size), and adult body size, and between thermal physiological response variables ($CT_{min}$, $T_b$, $T_{pref}$ and $CT_{max}$) and environmental predictors (bio1, bio10, bio11, and $T_e$). For the PGLS analysis (and for the SLOUCH analysis, described below) between $T_{pref}$ and environmental predictors we did not include *Sceloporus graciosus*, which represents one outlier point (as this species inhabits an extremely cold habitat). If we include *S. graciosus*, the PGLS regression is significant for oviparous species, but the strength of the relationship is weak ($y = 0.094x + 32.78$, $P = 0.03$).

*Assessing environmental predictors of parity mode evolution.* Given the strong conceptual framework linking the evolution of viviparity to cold environments, we tested whether changes in the thermal environment were strong predictors of parity mode shifts using two approaches. We tested for the evolutionary covariation between the thermal environment and reproductive mode (oviparous vs. viviparous) using the threshold model[115,116] using *threshBayes* function in the phytools (ver. 0.6.99) R package[107] and with phylogenetic logistic regression using *phyloglm* function in the phylolm (ver. 2.6.2) R package[117]. The threshold model is used to test for evolutionary covariation between continuous and discrete traits[116]. Under the threshold model, a discrete character (i.e., oviparity or viviparity) evolves as a function of a continuously varying feature (termed "liability"). When the value of "liability" crosses a certain threshold, the state of the discrete character evolves (i.e., a transition from oviparity to viviparity occurs)[115,116]. We ran threshBayes for $1.0 \times 10^6$ generations, sampling every 100 generations, and discarding the first 200 K generations as burn-in. We ran separate analyses for mean annual temperature (bio1), mean temperature of the coldest quarter (bio10), and mean temperature of the warmest quarter (bio11). The phylogenetic logistic regression is used to test if dependent variables (binary traits that switch between 0 and 1) are predicted by independent variables (continuous or discrete)[54]. So, we coded oviparity and viviparity using 0 and 1, respectively. As above, our predictor variables in these analyses were bio1, bio10, and bio11.

*Stochastic linear Ornstein-Uhlenbeck models.* Our OUwie analyses revealed reductions in the phenotypic optimum (θ parameter) for thermal traits in viviparous lizards (see Results and discussion). Yet, it is unclear whether reductions in thermal physiology reflect adaptation to cool environments (given, for example, the greater representation of viviparous lineages at high elevation[20,21]) or, instead, reflect energetic adjustments for life-history evolution (hypothesis i in Table 1a), which could be readily co-opted for life in cold environments. If cool-adapted physiology reflects adaptation to cool environments, there should be a strong evolutionary association between the local thermal environment and thermal physiology. However, if cool-adapted physiology more reflects a number of potential trade-offs like life-history energetics, then we expect viviparous species to exhibit more cool-adapted physiology than oviparous species regardless of ambient conditions, which should weaken the evolutionary relationship between the local thermal environment and thermal physiology.

To test these ideas, we used the SLOUCH model of ref. [118], which simultaneously estimates an "evolutionary regression" and an "optimal regression" in an OU

framework. The evolutionary regression describes the observed relationship between climatic predictors (mean annual temperature (bio1), mean temperature of the warmest quarter (bio10), and mean temperature of the coldest quarter (bio11)) and physiological response variables ($CT_{min}$, $T_b$, $T_{pref}$ and $CT_{max}$), while accounting for the relatedness among species. The estimated "optimal regression", by contrast, describes the relationship between these variables predicted under an OU model, and assumes adaptation of the response variables to the predictor variables. In addition to the regressions, the model permits the estimation of phylogenetic half-life ($t_{1/2}$) and rate of adaptation (α). Phylogenetic half-life represents the amount of time required for viviparous or oviparous lineages to get halfway to their thermal physiological optimum. So, a short $t_{1/2}$ (relative to the length of tree) indicates the phylogenetic signal degrades at a rapid pace. By contrast, a $t_{1/2}$ approaching (or exceeding) the length of the tree, indicates a strong phylogenetic signal. A rate of adaptation close to 0 represents a very slow physiological adaptation to thermal predictors, whereas α values higher than 100 (or approaching ∞) indicate instantaneous physiological adaptation to thermal predictors[119].

The similarity between the evolutionary and optimal regressions is supported when $t_{1/2}$ is close to 0, which would indicate that transitions in the thermal environment are rapidly coupled with changes in thermal physiology. Differences in the slopes of these relationships, by contrast, are supported when the phylogenetic half-life ($t_{1/2}$) of the model is bounded away from zero, implying phylogenetic inertia, or a lag in physiological adaptation to the thermal environment. Such lags are consistent with the Bogert effect, in which behavioral preferences disrupt physiological adaptation to prevailing environmental conditions[120–122]. Under this scenario, shifts in the thermal environment are predicted to be weakly associated with shifts in thermal physiology. To run the analyses, we simultaneously estimated the evolutionary regression, optimal regression, and $t_{1/2}$ for each thermal physiological trait ($CT_{min}$, $T_b$, $T_{pref}$ and $CT_{max}$) of phrynosomatid lizards, with respect to their thermal environment (bio1, bio10, and bio11) in an OU modeling framework using the *slouch.fit* function the R package SLOUCH (ver. 2.1.2)[118].

*Graphics.* Figures 1b, 2, and 3 were generated using the R package ggplot2 (ver. 3.2.1)[123], and edited using Adobe Illustrator.

**Reporting summary.** Further information on research design is available in the Nature Research Reporting Summary linked to this article.

## Data availability
The behavioral, life history, morphological, and physiological data, as well as GenBank accession numbers, compiled or generated in this study, are provided in the Supplementary Data files. Source data are provided with this paper.

## Code availability
The code used to generate all analyses in this study is provided in the Supplementary Data 6.

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

## Acknowledgements

S.F.D.-G., D.M.A.-M., A.B.-V., F.J.M.-N., and R.S.-B. are thankful to Posgrado en Ciencias Biológicas, Instituto de Biología-UNAM and Consejo Nacional de Ciencia y Tecnología (CONACyT) for the scholarships CVU 478292, 412744, 545201, 775920, and 774550, respectively. This research was supported by projects PAPIIT-UNAM IN210116 and IN212119 awarded to F.R.M.-C., 61866 from the Templeton Foundation awarded to M.M.M., and CONACyT PDCPN 2015-1319 awarded to P.G.-T.

## Author contributions

S.F.D.-G., F.R.M.-C., N.L.M.-M., M.E.O., and M.M.M. design the study. S.F.D.-G., P.G.-T., F.R.M.-C., D.M.A.-M., A.B.-M., H.G., R.A.L.-R., C.A.M.-M., F.J.M.-N., R.S.-B., J.H.V.-V., and G.A.W.-P. performed field and laboratory work to obtain the physiological, morphological, and life-history data. S.F.D.-G., N.L.M.-M., and A.B.-V. built the ultra-metric tree. S.F.D.-G. and M.M.M. analyzed the data and drafted the manuscript. All authors contributed to subsequent revisions.

## Competing interests

The authors declare no competing interests.

## Additional information

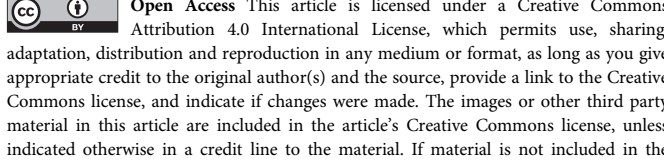

