## [Peer Review File · Nature Communications]

Exceptional parallelisms characterize the evolutionary transition to live birth in phrynosomatid lizardsReviewers' Comments:

Reviewer #1:

Remarks to the Author:

These papers should be cited:

Albuquerque, R. L., and T. Garland, Jr.. 2020. Phylogenetic analysis of maximal oxygen consumption during exercise ($VO_{2,max}$) and ecological correlates among lizard species. *Journal of Experimental Biology* 223:online early.

Clobert, J., T. Garland, Jr., and R. Barbault. 1998. The evolution of demographic tactics in lizards: a test of some hypotheses concerning life history evolution. *Journal of Evolutionary Biology* 11:329–364.

Hodges, W. L. 2004. Evolution of viviparity in horned lizards (*Phrynosoma*): testing the cold-climate hypothesis. *Journal of Evolutionary Biology* 17:1230–1237.

Perhaps also check this for similar energetic arguments regarding nocturnality:

Autumn, K., D. Jindrich, D. DeNardo, and R. Mueller. 1999. Locomotor performance at low temperature and the evolution of nocturnality in geckos. *Evolution* 53:580–599.

Figure 2: Add the key, symbols, directly on the figures to make it easier for readers.

Figure 2 legend: Say whether analyses indicate a difference or not. Refer to where those results are presented, e.g., a table.

Figure 3: Add the key, symbols, directly on the figures to make it easier for readers.

368: typo gravis

436 Lifespan needs a lot more explanation. Is this "physiological" e.g., as determined for animals kept under good conditions in the lab, or is it for animals in the wild, where most die from predation way shorter than they might live. What does "maximum" mean here and what are the implications of using whatever you used?

444 Estimation of metabolic rate is not acceptable for this sort of analysis. It is just derived from other traits in the analysis. This is a fatal flaw. Metabolic rate needs to be dropped. Otherwise, you need to obtain real data on it.

Reviewer #2:

Remarks to the Author:

Viviparity (live-bearing reproduction) has become a subject of intensive investigation by biologists from a wide variety of standpoints. Given that this pattern has originated in more than 100 separate clades of squamate reptiles, most viviparity research over the past few decades has been conducted on lizards and snakes.

This manuscript focuses on the large lizard family Phrynosomatidae of North and Central America, a group whose viviparity has received particular attention. The manuscript draws on Genbank data to reconstruct a phylogeny of the family and analyzes phenotypic factors associated with each of the

evolutionary transitions from oviparity to viviparity. Through a sophisticated analysis of a large database of information, the study demonstrates that parallel transitions to viviparity are associated with reductions in mass-specific metabolic rate and mass-specific production, factors that reflect decreases in fecundity and thermal physiology. Accordingly, the paper's analysis shows that evolution of viviparity is associated with decreases in energy allocated into maintenance and reproduction. These results entirely upend conventional claims of how cold climate selects for viviparity in reptiles.

The findings in this manuscript are astonishing; they require us to rethink conventional wisdom about how viviparity evolves. Past analyses based on simple correlations have concluded that viviparity is an adaptation to cold climate. In contrast, the present study offers a novel causal relationship in which thermal and metabolic adjustments to cooler climate facilitated the widespread colonization of cold environments. Significance of the analysis extends far beyond phrynosomatid lizards to reptiles in general, along with viviparous fishes, amphibians, and mammals.

A single suggestion for modification follows. Reproduction in *Sceloporus goldmani* is labeled in the manuscript as "oviparous" based on an observation of a female said to have laid two eggs. While information in the literature is scanty, at least two reputable sources have provided definitive evidence of viviparity (Smith & Hall, 1974; Carbajal-Márquez & Quintero-Díaz, 2017); these include observations of birth in captivity, and discovery of well-developed fetuses in the holotype and a paratype of the species. Given the paper's phylogeny, this species would seem to represent an additional (sixth) origin of viviparity. The authors may have reason to question the cited literature sources. If not, options include redoing the analysis, or omitting *goldmani* on the grounds of discrepant information.

This paper is a game-changer; it will surely prove highly influential in affecting how we think about, and analyze, the evolution of viviparity and other reproductive patterns. It certainly warrants publication in a prominent venue, and is certainly appropriate for *Nature*.

Smith, H. M., & Hall, W. P. (1974). Contributions to the concepts of reproductive cycles and the systematics of the *scalaris* group of the lizard genus *Sceloporus*. *Great Basin Naturalist*, 97-104.

Carbajal-Márquez, R.A. & G.E. Quintero-Díaz (2017). Natural history of *Sceloporus goldmani* (Squamata: Phrynosomatidae) in its southern distribution. *Herpetology Notes* 10: 161-167.

We thank both of the reviewers very much for all their comments. In this new version we have addressed all the points raised by the reviewer, as detailed below.

Reviewer 1

Comment:

These papers should be cited:

Albuquerque, R. L., and T. Garland, Jr.. 2020. Phylogenetic analysis of maximal oxygen consumption during exercise ($\dot{V}O_{2,max}$) and ecological correlates among lizard species. *Journal of Experimental Biology* 223:online early.

Clobert, J., T. Garland, Jr., and R. Barbault. 1998. The evolution of demographic tactics in lizards: a test of some hypotheses concerning life history evolution. *Journal of Evolutionary Biology* 11:329–364.

Hodges, W. L. 2004. Evolution of viviparity in horned lizards (*Phrynosoma*): testing the cold-climate hypothesis. *Journal of Evolutionary Biology* 17:1230–1237.

Perhaps also check this for similar energetic arguments regarding nocturnality:

Autumn, K., D. Jindrich, D. DeNardo, and R. Mueller. 1999. Locomotor performance at low temperature and the evolution of nocturnality in geckos. *Evolution* 53:580–599.

Response: We appreciate the reviewer's suggestions, and have incorporated them into the text.

Comment:

Figure 2: Add the key, symbols, directly on the figures to make it easier for readers.

Response: We agree with the reviewer's point and made the suggested change.

Comment:

Figure 2 legend: Say whether analyses indicate a difference or not. Refer to where those results are presented, e.g., a table.

Response: Changed.

Comment:

Figure 3: Add the key, symbols, directly on the figures to make it easier for readers.

Response: Done.

Comment:

368: typo gravis

Response: Corrected.

Comment:

436 Lifespan needs a lot more explanation. Is this "physiological" e.g., as determined for

animals kept under good conditions in the lab, or is it for animals in the wild, where most die from predation way shorter than they might live. What does "maximum" mean here and what are the implications of using whatever you used?

Response: We refer to maximum lifespan in natural conditions. Because lifespan in natural conditions is not commonly reported for phrynosomatid lizards, and because the few data available indicate that it varies among species (regardless of parity mode), we standardized production to one year make to data comparable among species. We provide additional explanation in our revision.

Comment:

444 Estimation of metabolic rate is not acceptable for this sort of analysis. It is just derived from other traits in the analysis. This is a fatal flaw. Metabolic rate needs to be dropped. Otherwise, you need to obtain real data on it.

Response:

The reviewer's concern, we believe, is that they find predicting metabolic rate using metabolic scaling equations inappropriate and prefer, instead, that the metric be empirically derived. We appreciate the reviewer's perspective, and certainly appreciate the value of empirically derived measures. Nonetheless, we respectfully disagree with their perspective and find it inconsistent with the field. We explain our reasoning below.

The metabolic theory of ecology establishes that individual metabolic rate (I) scales with mass and increase exponentially with temperature^{1,2}.

$$I = i_0 M^{3/4} e^{-E/kT}$$

Such that i_0 is a normalization constant, M is mean body mass (g) of individuals, e = Euler's number, E = activation energy, k = Boltzmann's constant, and T = body temperature (in Kelvin)². To tailor these equations to lizards, we used $i_0 = \ln(20.3)$ (normalization constant for reptiles) and an E value = 0.63 (activation energy excluding endotherms in hibernation and torpor)².

The universal statistical validity of this fundamental equation of the metabolic theory of ecology has been (rightfully) debated^{4,5,6,7,8,9}. However, this equation is a useful (and very widely applied) approximation for metabolic rate across species for macrophysiological studies^{1,3,6,10,11}. For example, Dillon et al. (2010, *Nature* [reference 3 below]) used the same approach to estimate vital rates across a deep slice of ectotherm animal diversity, allowing for a comparison of how metabolism will be impacted by climate warming.

Of course, the slope and intercept of the metabolic scaling relationship can (and does) vary across animal lineages. We selected values that reflected our specific lineage (lizards). A reasonable counterargument could be that within-reptile evolution alters this relationship. Yet, a recent macroevolutionary analysis has revealed that there are no major evolutionary transitions in the slope and intercept of the scaling relationship in reptiles¹². In other words,

these values are fairly robust across reptiles. Correspondingly, empirically derived and mathematically estimated metabolic rates tend to align.

Shifts in the slope and intercept of the scaling relationship among reptiles would not alter our predicted metabolic rates substantially and certainly would not alter our general finding (lower mass-specific MR in viviparous lineages), as this feature so clearly reflects the substantially cooler thermal physiology of those species. Further, and perhaps more saliently, our study centers around unpacking the evolutionary pathways associated with the reduction in mass-specific metabolic rate (MSMR), rather than the reduction in MSMR itself (this latter observation being already generally recognized in the literature, using both experimental and mathematical estimates of metabolic rate^{13,14}). We believe that our approach is robust and it is consistent with top papers in the field.

Importantly, experimental estimation of the metabolic rate would not have addressed the specific question we had. Studies that provide empirical estimates of metabolic rate using respirometry approaches do so by keeping the experimental temperature constant. Doing so allows researchers to compare metabolic rate among different individuals, populations, and species across shared temperature regimes. Here, we took a different tack by predicting the minimum metabolic demands of an organism given its known body size (from our data) and its known field-measured body temperature (also empirically measured). This allowed us to predict how MRs should vary based on the observed ecology of the species, as opposed to comparing the metabolic demands across shared thermal conditions. In other words, we tailored the equations to the observed ecology of the species to determine the energetic implications of those features. Through our approach, we were able to unpack the significance of the evolutionary reductions in thermal physiology that we observed in viviparous lineages.

We recognize, nonetheless, that we could have enhanced the clarity and utility of our approach. In our revision we enhance the detail regarding the reptile-specific constants that we used in our analysis (as these tailored the equations to our system). We also provide greater context surrounding the utility of metabolic equations, acknowledge the debate surrounding their use, and better describe the relevance of our approach for our specific research questions.

References

1. Gillooly, J. F., Brown, J. H., West, G. B., Savage, V. M. & Charnov, E. L. Effects of size and temperature on metabolic rate. *Science* **293**, 2248–2251 (2001).
2. Brown, J. H., Gillooly, J. F., Allen, A. P., Savage, V. M. & West, G. B. Toward a metabolic theory of ecology. *Ecology* **85**, 1771–1789 (2004).
3. Dillon, M. E., Wang, G. & Huey, R. B. Global metabolic impacts of recent climate warming. *Nature* **467**, 704–706 (2010).
4. Clarke, A. & Fraser, K. P. P. Why does metabolism scale with temperature? *Funct. Ecol.* **18**, 243–251 (2004).

5. Clarke, A. Temperature and the metabolic theory of ecology. *Funct. Ecol.* **20**, 405–412 (2006).
6. Gillooly, J. F. *et al.* Response to Clarke and Fraser: Effects of temperature on metabolic rate. *Funct. Ecol.* **20**, 400–404 (2006).
7. O’Connor, M. P. *et al.* Reconsidering the mechanistic basis of the metabolic theory of ecology. *Oikos* **116**, 1058–1072 (2007).
8. Downs, C. J., Hayes, J. P. & Tracy, C. R. Scaling metabolic rate with body mass and inverse body temperature: A test of the Arrhenius fractal supply model. *Funct. Ecol.* **22**, 239–244 (2008).
9. Isaac, N. J. B. & Carbone, C. Why are metabolic scaling exponents so controversial? Quantifying variance and testing hypotheses. *Ecol. Lett.* **13**, 728–735 (2010).
10. Del Rio, C. M. Metabolic theory or metabolic models? *Trends Ecol. Evol.* **23**, 256–260 (2008).
11. Brown, J. H. & Sibly, R. M. The metabolic theory of ecology and its central equation. In *Metabolic Ecology: A scaling approach* (eds. Sibly, R. M., Brown, J. H. & Kodric-Brown, A.) (Wiley & Sons, Ltd, 2012).
12. Uyeda, J. C., Pennell, M. W., Miller, E. T., Maia, R., and McClain, C. R. The evolution of energetic scaling across the vertebrate tree of life. *American Naturalist* **190**, 185-199 (2017).
13. Zhang, L., Guo, K., Zhang, G. Z., Lin, L. H. & Ji, X. Evolutionary transitions in body plan and reproductive mode alter maintenance metabolism in squamates. *BMC Evol. Biol.* **18**, 45 (2018).
14. Healy, K., Ezard, T. H. G., Jones, O. R., Salguero-Gómez, R. & Buckley, Y. M. Animal life history is shaped by the pace of life and the distribution of age-specific mortality and reproduction. *Nat. Ecol. Evol.* **3**, 1217–1224 (2019).

Reviewer 2

Comment:

Viviparity (live-bearing reproduction) has become a subject of intensive investigation by biologists from a wide variety of standpoints. Given that this pattern has originated in more than 100 separate clades of squamate reptiles, most viviparity research over the past few decades has been conducted on lizards and snakes.

This manuscript focuses on the large lizard family Phrynosomatidae of North and Central America, a group whose viviparity has received particular attention. The manuscript draws on Genbank data to reconstruct a phylogeny of the family and analyzes phenotypic factors associated with each of the evolutionary transitions from oviparity to viviparity. Through a sophisticated analysis of a large database of information, the study demonstrates that

parallel transitions to viviparity are associated with reductions in mass-specific metabolic rate and mass-specific production, factors that reflect decreases in fecundity and thermal physiology. Accordingly, the paper's analysis shows that evolution of viviparity is associated with decreases in energy allocated into maintenance and reproduction. These results entirely upend conventional claims of how cold climate selects for viviparity in reptiles.

The findings in this manuscript are astonishing; they require us to rethink conventional wisdom about how viviparity evolves. Past analyses based on simple correlations have concluded that viviparity is an adaptation to cold climate. In contrast, the present study offers a novel causal relationship in which thermal and metabolic adjustments to cooler climate facilitated the widespread colonization of cold environments. Significance of the analysis extends far beyond phrynosomatid lizards to reptiles in general, along with viviparous fishes, amphibians, and mammals.

This paper is a game-changer; it will surely prove highly influential in affecting how we think about, and analyze, the evolution of viviparity and other reproductive patterns. It certainly warrants publication in a prominent venue, and is certainly appropriate for Nature.

Response: We are humbled by the reviewer's words. Thank you so much.

Comment:

A single suggestion for modification follows. Reproduction in *Sceloporus goldmani* is labeled in the manuscript as "oviparous" based on an observation of a female said to have laid two eggs. While information in the literature is scanty, at least two reputable sources have provided definitive evidence of viviparity (Smith & Hall, 1974; Carbajal-Márquez & Quintero-Díaz, 2017); these include observations of birth in captivity, and discovery of well-developed fetuses in the holotype and a paratype of the species. Given the paper's phylogeny, this species would seem to represent an additional (sixth) origin of viviparity. The authors may have reason to question the cited literature sources. If not, options include redoing the analysis, or omitting *goldmani* on the grounds of discrepant information.

Smith, H. M., & Hall, W. P. (1974). Contributions to the concepts of reproductive cycles and the systematics of the scalaris group of the lizard genus *Sceloporus*. *Great Basin Naturalist*, 97-104.

Carbajal-Márquez, R.A. & G.E. Quintero-Díaz (2017). Natural history of *Sceloporus goldmani* (Squamata: Phrynosomatidae) in its southern distribution. *Herpetology Notes* 10: 161-167.

Response: We greatly appreciate the reviewer's perspective. Given there is controversy surrounding the reproductive biology of the different populations assigned to *Sceloporus goldmani* we have opted to include an additional set of analyses in which we treat this species as viviparous and examine whether doing so affects our results. Of course, considering *S. goldmani* viviparous raises the number of independent transitions in

phrynosomatids to 6, rather than 5, because this species is well nested within an oviparous clade. The results of our model-fitting analyses, however, are unchanged by considering this species to be viviparous. We provide these additional analyses in our revision.

Reviewers' Comments:

Reviewer #3:

Remarks to the Author:

I was asked to address a point raised by the reviewer, who objected to the estimates of metabolic rate (rather than empirical data), which came from using a general equation from The Metabolic Theory of Ecology. Given that empirical measurements are available for only a few species, I accept the arguments of the author to the effect that the MTE equation is an adequate first approximation. Still it would be good to check predictions against empirical data, as MR has surely been measured for some species (*Sceloporus occidentalis*, *Uta stansburiana*-- check Bennett and Dawson's chapter in B of R for a start).

Overall, I think this is a very interesting, impressive, and broadly conceived paper. It merges life history, thermal biology, behavior, climate, and evolutionary history. Although it is not the approach I myself would have taken (I'd consider using a biophysical model with a full life history (Ma et al., 2018, see below) but this approach brings in an evolutionary perspective that is currently missing from biophysical approaches.

I must confess that I don't understand everything, especially the type of phylogenetic analyses used here (SLOUCH model). I find the use of "optimal" here to be confusing, as SLOUCH's "optimal" seems to bear little resemblance to the traditional use of "optimal" in evolutionary studies. Also, SLOUCH assumes a OU framework. Is there any evidence that this model applies to these lizards? Note: I suspect I'm not alone in being ignorant about SLOUCH.

I'm not convinced by the conclusion in the Abstract (lines 40-43). If viviparity is not "...an adaptation to "cold climates per se," but instead "facilitates...colonization of cooler environments," then what favors the initial evolution of viviparity prior to that colonization? The classic "beauty" of the cold-climate hypothesis was that it allowed for intermediates states (egg retention in cold climates, leading eventually to viviparity).

Thus, I don't see what favors the initial evolution of viviparity (intermediate steps) in a warm environment. Moreover, if a lower preferred body temperature is a key step in reducing metabolic rate in proto-viviparous species in a warm environment, then a clear disadvantage of that low T_b is that it must reduce potential activity time, at least in warm seasons. [See Sinervo et al. 2010.] So I must be missing something in the logic.

One other issue is the use of field T_b to estimate metabolic rate. I see two problems here. First, T_b is apparently sampled from active lizards, but most lizards are active only a few hours per day. Thus, if you are trying to calculate metabolic rate, you really need time-series data on T_b , including T_b during inactive periods. I realize these won't be available for these species. This is one reason why I'd use a biophysical approach -- one could couple NicheMapR and a behavioral model to roughly estimate metabolic rate and energetic costs. At a minimum you could speculate on whether your conclusions would likely hold if you had hourly T_b data over the entire reproductive season. Second, I assume you plugged mean T_b and mean mass into these calculations. This will introduce a minor error, given the nonlinearity of $E \times T$ relationships and $E \times M$ relationships. See this Savage paper.

Savage, V. M. 2004. Improved approximations to scaling relationships for species, populations, and ecosystems across latitudinal and elevational gradients. *Journal of Theoretical Biology* 227:525-534.

I encourage you to contrast your findings with that of Ma et al. (2018). They used a biophysical model and estimated that the total cost of development is lower for viviparous species than for oviparous embryos. They concluded that the main advantage of viviparity is that it shortens development time in cold environments. There's some overlap and discrepancies in perspectives between these two papers.

Similarly, Beuchat and Vleck (1990) measured metabolism of gravid *Sceloporus jarrovi* (viviparous), and found that the metabolism of gravid females during pregnancy is higher than expected based on maternal body mass.

[As an aside, I'd encourage you to look at the doubly-labeled water literature. Ken Nagy measured field metabolic rates on many lizards (including phrynosomatids), but I don't recall whether he studied any viviparous species. If so, this would be an interesting comparison.]

Again, I like the paper. It is long and complex, and some of it is well outside my expertise.

Miscellaneous comments. Key ones are preceded by ***. [Note that some issues are eventually addressed in the Methods section.]

26 Delete "strong"

27 Rewrite sentence as syntax is off. "origins" literally refers here to life history pathways, but you want it to refer to innovation.

31 add "in" in front of mass-specific production

34 But won't these changes slow development, and thus the length of time the female is carrying embryos?

35 What is a "thermal habitat"

38 "track" seems inappropriate, given a million-year lag!

39 Unclear.

41-3 Unclear.'

54 Squamates? or other taxa, too?

96... Many traits here. Were the analyses also done with mean values, thus assuming no error in measurement?

107 I assume you mean "mean annual AIR temperature" here. Given that some lizards have broad geographic ranges, how did you assign single temperatures to a species? Or did you use only the particular locality that you studied.

113 Syntax. "To...results, we used evolutionary regressions to explore..."

119 Rewrite

162 *** That is clear, but is age of first reproduction independent of body size? And is age of first reproduction independent of parity? Age of first reproduction is a dominant contributor to fitness (Cole 1954, Stearns, etc.)

165 *** Personally, I doubt that there's global stabilizing selection on "quality and quantity of offspring." Clutch and egg size varies dramatically among populations in *Sceloporus occidentalis* and from clutch to clutch in *Uta stansburiana* (old work by Sinervo). Sinervo showed that optimal egg size varied among clutches, as I recall.

179 Rewrite. "..., females increase in mass but reduce Tb

183 Here you use "optimal temperature" in a very different way from the rest of the text. This is confusing to me.

Interesting idea, but the mass increases because of embryos, which will probably have high mass-specific rates because they are small and developing. The impact of this will be small, however.

187 Check Beuchat and Elnor. Interesting idea.

189 You are using instantaneous values for metabolic rate, but given that gestation time is much longer in viviparous species, what about cumulative metabolic rates (over entire reproductive period).

You could do a quick analysis to check to see whether lower body temperature do compensate for longer gestations. I'm not convinced at a drop in Tb of a few degrees "compensates" for the much longer pregnancy.

196 Please explain your logic here. Are you referring to body or local temperature?

201 When you present results, you focus on "optimal" values not actual values. The latter are mor meaningful to me, perhaps because I'm not understanding optimal in this context.

230 I don't follow.

231 Shouldn't you use operative temperature here, as air temperature can be a misleading indicator of the thermal environment? See work by Porter, Kearney, Sunday, and many others.

***Also, consider looking at seasonality (difference between hot and cold seasons). This will provide crude indication of the length of the active season (inverse relationship) I suspect.

248 I realize this citation came from a reviewer, but the sentence is unclear to me. I could imagine many things that could be called fitness-related behaviors. If you are referring to the co-evolution of Tpref and Topt, see Huey & Bennett Evolution 1987

Fig. 3 Interesting patterns. How do you account for the negative slope (though not significant) for viviparous lizards in 3c?

In 3C, there are two outlier points: the left most oviparous, and the lowest viviparous. I was taught that comparisons of regressions should use shared ranges of the x variable, where possible. If you toss out the left-most MAT, which will have high leverage, is the correlation still significant? If you toss out the one viviparous species with Tpref ~ 28 °C, what happens to the correlation?

I'm dubious about that low Tpref. None of the Tpref in table S1 are that low. Are you confident that is a valid estimate?

253, 254 $p = 0$? Report P values to more digits.

259 Something is wrong in this sentence.

269 Operative temperatures would be more robust here.

271 I think you are saying that viviparity evolves first and then this allows species to move into cooler environments. If so, what favors the evolution of viviparity? Am I missing something.

274 Doesn't the idea go back to Mell, or Weekes, or Sergeev?

296 I'm not convinced that viviparous females do have a reduced "mass-specific energetic burden."

Yes, they will on an instantaneous basic, but you need time series of Tb (including inactive periods) and a factor for the much longer gestation. In other words, metabolic rate integrated over the entire reproductive period. Cumulative MR might still be lower, but that is unknown at present.

357 Supplementary Data 2 is not in the "Supplementary information." I'd really like to see the input data, sample sizes, sources (literature vs. original). The only data on Tb, Tpref, etc. are for about $N \sim 26$ species (male vs female comparisons) in Supplementary Table 1.

362 "fails TO right"

365 your new data collection? Sentence unclear

367 Unclear on environmental variables. If a specie was known from 100 localities, did you compute weighted values of mean temperature, for example, or from only the locality where Tb data were collected? I think the latter.

372 See Beuchat & Ellner, below

373 Why exclude data on pregnant females, as aren't these the focus of your study? I suspect this is because the lab data aren't available. But if females drop Tp when pregnant, won't this just strengthen the pattern you observe? If so, you should consider saying that your estimates are thus conservative.

375 Did you use paired t-tests?

375 This is not universally the case. Compare Lailvaux (2007) vs Huey and Pianka (2007)
379 Now we understand the decision in line 373. But are these data completely lacking?
388 Ref 41 looked at 1 population, but the wording here implies multiple species were validated.
393 1-5 days of sampling is hardly ideal, as Tb on those days might reflect weather or season, etc.
These lizards "exhibit surface activity" for many months of the year, if not all year in some localities.
How can a 1-5 day snapshot adequately characterize a species? [e.g., look at seasonal variation in Tb
in Kalahari lizards, Huey and Pianka 1977.] Furthermore, how did you pick localities? Did you visit
localities where physiological data were taken?
You could use NicheMapR to get a more comprehensive portrait of the thermal environment.

395 Check Beuchat.

395 *** I assume that the studies here compared *active* body temperatures of viviparous vs.
oviparous taxa. But lizards are active only for part of the day, and active Tb thus fail to capture the
24-h dynamics of Tb. Has anyone used telemetry on viviparous vs oviparous lizards in the same
habitat?

398 Wouldn't Tpref be the key metric here?

403... The results will vary seasonally in most species.

420 Impressive. What fraction of these data are new versus taken from the literature?

427 As above, these are air temperatures.

438 What do you do with species with broad geographic ranges, which may have major variation in N
eggs per clutch,

N clutches per year, and size of eggs? (see Sinervo's old work with *Sceloporus*).

441 Some or perhaps many oviparous species have multiple clutches, but not all.

447 I'm surprised viviparous species don't live longer. How broad is the sampling in ref 45? You are
missing one key element, namely the age of first reproduction. Given their often-cold habitats,
viviparous species likely have delayed onset of reproduction.

484 Why transform to integer value.

489 "neonate" or "egg" mass in oviparous species?

534 Complex sentence. Perhaps start with "We performed PGLS...nlme to depict the evolutionary
relationship between ..."

585 Move the citation for ref 13! I thought you were raising M to the 13th power!

Supplementary Information. I don't see a table with all of the data (field Tb, Tpref, size, ect. and the
data sources.

Beuchat, C. A. 1986. Reproductive influences on the thermoregulatory behavior of a live-bearing
lizard. *Copeia* 1986:971-979.

Beuchat, C. A., and S. Ellner. 1987. A quantitative test of life history theory: thermoregulation by a
viviparous lizard. *Ecological Monographs* 57:45-60.

Beuchat, C.A. and D. Vleck. 1990. Metabolic consequences of viviparity in a lizard, *Sceloporus jarrovi*.
Physiol. Zool. 63:555-570.

Ma, L., Buckley, L.B., Huey, R.B., and Du, W.-G. 2018 A global test of the cold-climate hypothesis for
the evolution of viviparity in squamate reptiles. *Global Ecol Biogr* 2018:1-11.

Ray Huey

Reviewer #4:

None

Reviewer #5:

Remarks to the Author:

There is a lot going on in this paper and in general there seems to be some striking differences in a range of characteristics between viviparous and oviparous lizards, which are pretty exciting. But the impact of the study is hard to judge at the moment because the presentation is complex and fragmented with predictions being only vaguely linked with hypotheses, which in some places don't seem to be articulated at all. The analyses are similarly hard to untangle, with a whole battery of tests being thrown at the data, but with no clear structure as to what the specific aim might be for each analysis set and how it relates to the overarching goals of the study. This could probably be remedied by having clearly argued hypotheses that then lead to a logical set of analyses that a reader can intuitively understand and relate back to those hypotheses. At the moment, however, the paper opens with a very general and very short 'background' paragraph and then gets quickly bogged down in the next paragraph with vague discussions of mass-specific metabolism and reproductive outputs that somehow lead to the set of predictions listed in Table 1. What the specific hypotheses might be that produce these predictions aren't really outlined, nor is it really clear how these predictions relate to the origin of viviparity briefly mentioned in the opening paragraph. Predation and life history are then mentioned at some point, too. The authors are heavily constrained here on the concise format of Nature Communications, but its even more vital for this format that a study is carefully structured to convey the study's goals and findings effectively.

It seems to me that the study has two aims. The first is related to the title of the paper and is about quantifying the various characteristics that might have converged in viviparous species. The second seems to be about testing the putative adaptive origins of viviparity itself. This isn't presented until very late in the paper, despite being mentioned near the start of the opening paragraph. And logically it seems that this 'why did viviparity evolve?' should start the study and then be followed by 'once it evolved, does it then lead to predictable adaptive changes in other characteristics?'

The origin of viviparity seems to have been tied in the literature to the invasion of cold climates. All this information is presented in a single sentence in the opening paragraph and this probably needs to be outlined in more detail right at the outset so readers can follow. The authors have rightly taken the opportunity to test this idea in these lizards. The threshold analysis seems to be a relevant method for doing so, but there's no report of whether the effects computed by this analysis (I'm assuming the r values are effect sizes) are or are not statistically distinguishable from zero. Either these effects need to be reported with confidence intervals, or a model selection approach is needed to pit alternative models/outcomes against each other (like the BM vs OU models in the OUwie analyses). If the threshold analysis doesn't allow this, then a phylogenetic logistic regression would at least be able to provide this type of statistical benchmark. Here, if cold environments are associated with the shift to live birth, then you'd see the switch point in the trend line of the associated graph to the analysis. If it isn't — and it doesn't appear to be based on the reconstructions of temperature in the SI — then the analyses would be able to compute the effect (or lack thereof) and an associated confidence interval.

Given cold environments don't seem to be associated with the origin of viviparity, are there no other hypotheses? It seems there should be an opportunity here to not only refute a prevailing hypothesis in the literature, but also to test other possible selection pressures for viviparity too. This would really push the paper up in impact.

On the other aim of the study — the parallelisms highlighted in the title of the paper — I'm not sure the analyses applied by the authors really do justice to this. The optima shown in the figure of the main text is simply the phylogenetic mean of the characteristics in viviparous and oviparous species, but it doesn't capture the variance. Note, this isn't the variance in estimated optima across the 500 stochastic reconstructions of ancestor states (which is shown), but the variance of the individual optima computed for each of those reconstructions themselves. This is very different and goes directly to the extent these differences between viviparity and oviparity are in fact statistically distinguishable

from one another. That is, these plots (or ones in the SI) need to be show the upper and lower confidence intervals of each of these computed optima. It can still be a jitter plot or a frequency distribution of points corresponding to the upper and lower CI values, but this way it will be clear what the statistical effect really is.

Judging from the tables, it seems that OUwie is either identifying the multiple optima OU model as being the most supported or at least within 2 DAICs of an alliterative. But first, why are these single values? This has presumably been done 500 times across all the stochastic maps, and 500 times across a set of posterior trees, so why are these not mean AIC values or better yet, plotted as a frequency distribution so the overlap in support between the alternative models is clear. Second, why not just use the OU multiple peak model for all characteristics based on the fact its typically supported in most cases? It would then be possible to convert the phylogenetic half-life estimate into an alpha value (Hansen's original paper provides this equation) that provides a statistical measure of the rate of adaptation, as well as a sigma² value that measures the magnitude of stochasticity in the evolution process. These together would help show what I suspect is stabilising selection operating on these characteristics in viviparity species, and perhaps not in the other oviparity species. The point being, focussing on just theta (the putative optima of the characteristic) provides a very small picture of what might have been happening in these lizards.

More generally, it seems what the authors are really trying to do is document convergences among the five lineages of viviparity. The current analyses applied do not address this. The obvious approach would be to apply a phylogenetic PCA to identify how these characteristics are related to each other and which might be representative of orthogonal axes. And then use these representative characteristics or the PCA components themselves in a 'surface' analysis (the authors are probably aware of this R package already; if not, they can google it along with Ingram & Mahler and that should bring it up). This analysis will find where convergences have occurred across the entire phylogeny. Presumably these five lineages will be clearly identified as being convergent, but it will also reveal instances why they might not be... which would be interesting in itself. This would then be followed by something link Kevin Arbuckle's method (also available in R) to confirm the magnitude of convergences amongst viviparity species relative to the variance across the rest of the tree. Importantly this would provide confidence intervals as well, not just the magnitude of similarity.

Currently this paper relies purely on a graphical presentation of theta and some basic model selection of BM, a common OU optima for all species and an OU that assumes viviparity and oviparity differ in optima. This latter analysis suggests convergence — or parrallism as the authors' put it — but it doesn't fully reveal it statistically. Again the OUwie analyse could still provide some resolution on changes in rates of adaptation to a viviparous lifestyle, and in particular the outcome of probable stabilising selection on these species (but not oviparous species).

I really couldn't figure out the relevance of the mass-specific metabolism angle. Regardless, it's inferred metabolism not measured metabolism. So basically the authors have just applied a sophisticate transform of body size. If the authors are attempting to test the extent to which viviparous and oviparous species conform to the Gillooly-Brown metabolism equation, then ok, but how it relates to the overarching goals of the study is not obvious. Instead it seems to be a complex set of analyses that doesn't have a clear take home message or really relate back to the origin of viviparity (or its subsequent adaptive significance).

This is an exciting data set, and there might be something big in it, but it's really hard to extract what it is from the current paper. I suspect a shift in focus of the analyses would help tremendously in providing this resolution.

Synopsis: We wanted to thank both reviewers for exceptionally clear and helpful feedback. The major comments centered around: (1) clarifying the conceptual context for our results (as articulated by both reviewers) and (2) performing additional analyses to quantify the degree of evolutionary convergence among phenotypic traits (comments by reviewer 5). We greatly appreciated this feedback, which we incorporated into our revision. First, we restructured our manuscript so as to better clarify the purpose of this study, and to better home our message. We believe the revision is clearer, and we appreciate the reviewer feedback, which encouraged us to more deeply consider how our arguments were presented. We also performed a series of additional analyses to assess convergence, which helped buttress our general conclusions. In addition, the reviewers provided a series of additional comments and suggestions, which we considered below, and which resulted in several additional analyses and edits. We found the feedback very helpful, and believe our revisions have resulted in a clearer and stronger manuscript. Thank you very much.

Reviewer 3

Comment:

I was asked to address a point raised by the reviewer, who objected to the estimates of metabolic rate (rather than empirical data), which came from using a general equation from The Metabolic Theory of Ecology. Given that empirical measurements are available for only a few species, I accept the arguments of the author to the effect that the MTE equation is an adequate first approximation. Still it would be good to check predictions against empirical data, as MR has surely been measured for some species (*Sceloporus occidentalis*, *Uta stansburiana*-- check Bennett and Dawson's chapter in B of R for a start).

Response:

We thank the reviewer for their feedback. We agree that it would be good to compare empirical data of MR among oviparous and viviparous phrynosomatids. We did a literature search for empirical estimates of MR in phrynosomatid lizards, and found measures for both oviparous and viviparous species, although the former were better represented in the literature (see Table 1 below). As the reviewer notes, MR scales exponentially with temperature¹. Correspondingly, when mass-specific MR is measured at shared thermal conditions, empirical estimates of MR result in similar values, independent of parity mode². For example, similarly sized individuals of *Phrynosoma cornutum* (oviparous) and *P. hernandesi* (viviparous) exhibit similar metabolic rates when they are exposed to shared experimental temperatures of 25°C, 35°C or 40°C³. The same pattern occurs in *Sceloporus aeneus* (oviparous) and *S. bicanthalis* (viviparous): individuals exhibit a similar mass-specific metabolic rate at experimental temperature of 33°C⁴. We found empirical data of resting mass-specific MR for 9 oviparous species and 3 viviparous species in phrynosomatids, in which MR was estimated under similar thermal conditions (see Table 1 below). These patterns are entirely consistent with our results: it is precisely the difference in activity temperature, rather than any size effects, which drives differences in (activity)

metabolic rate between parity modes. The relatively few species for which empirical data are available preclude a robust evolutionary analysis, but we nonetheless performed a phylogenetic ANOVA, and found a similar mass-specific MR among oviparous and viviparous phrynosomatids ($F_{1,10}=0.178, p=0.6$). Of course, the statistical power of this analysis is weak, but visually inspecting the data shows that mass-specific metabolic rates are similar between oviparous and viviparous phrynosomatids, when they are measured at similar temperatures (this makes sense and is consistent with our results). Of course, the whole point of laboratory-measured MRs, as the reviewer notes, is to understand the minimum energetic demand, given an organism's body size, and a set temperature (purposefully kept constant due to thermal effects on MR). Our analysis illustrates that species' field-measured body temperatures, which are lower in viviparous species, translate into the lower mass-specific MR during activity periods for those species. We clarify that our estimates of mass-specific MR relate to those during the period of activity, reflecting the field-active body temperatures of lizards estimated in the wild. In response to other comments below, we also added a comparison of mass-specific MR during periods of inactivity to better understand differences in metabolic demand among parity modes (described in detail below).

Table 1. Empirical mass-specific metabolic rate (msMR) for some phrynosomatid species. T refers to temperature ($^{\circ}\text{C}$) at which msMR was measured.

Species	Parity mode	msMR ($\text{cm}^3 \text{O}_2 \text{g}^{-1} \text{h}^{-1}$)	T	Ref.
Petrosaurus mearnsi	Oviparous	0.14 R	35	5
Phrynosoma cornutum	Oviparous	0.166 R	35	3
Phrynosoma mcallii	Oviparous	0.28 R	35	6
Sceloporus graciosus	Oviparous	0.299 R	35	7
Sceloporus olivaceus	Oviparous	0.191R	33	8
Sceloporus undulatus	Oviparous	0.237 R	33	9
Uta stansburiana	Oviparous	0.27 R	36	10
Sceloporus aeneus	Oviparous	0.376 R	33	4
Sceloporus merriami	Oviparous	0.15 R	34	11
Mean \pm SD		0.234 \pm 0.08	34.3 \pm 1.1	
Sceloporus bicanthalis	Viviparous	0.34 R	33	4
Sceloporus jarrovii	Viviparous	0.118 R	30	12
Phrynosoma hernandesi	Viviparous	0.171 R	35	3
Mean \pm SD		0.2097 \pm 0.12	32.5 \pm 3.8	

Comment:

Overall, I think this is a very interesting, impressive, and broadly conceived paper. It merges life history, thermal biology, behavior, climate, and evolutionary history. Although it is not the approach I myself would have taken (I'd consider using a biophysical model

with a full life history (Ma et al., 2018, see below) but this approach brings in an evolutionary perspective that is currently missing from biophysical approaches.

Response:

Thank you for your kind words. We additionally compared our results to that of Ma *et al.* (2018) in our revision.

Comment:

I must confess that I don't understand everything, especially the type of phylogenetic analyses used here (SLOUCH model). I find the use of "optimal" here to be confusing, as SLOUCH's "optimal" seems to bear little resemblance to the traditional use of "optimal" in evolutionary studies. Also, SLOUCH assumes a OU framework. Is there any evidence that this model applies to these lizards? Note: I suspect I'm not alone in being ignorant about SLOUCH.

Response:

We totally understand the reviewer's point, as the meaning of 'optimal' varies among contexts and different biological analyses. Phylogenetic comparative methods that examine patterns of adaptation rely on the Ornstein-Uhlenbeck (OU) model of adaptive evolution. SLOUCH incorporates the OU process to model trait evolution in the presence of one (or more) phenotypic 'optima' (denoted by the theta (θ) parameter)¹³. So, by 'optimum' we mean the evolutionary optimal phenotype based on an OU analysis of our trait data. The SLOUCH approach estimates an 'optimal' regression between thermal traits and thermal environments, which describes the anticipated trait~environment relationship under a scenario of complete adaptation. The 'evolutionary' regression, by contrast, illustrates the observed trait~environment relationship. Then, deviations in the slope between the 'evolutionary' and 'optimal' regressions illustrate phylogenetic lags in adaptation¹³. The terminology is a bit confusing because the word 'optimal' can be used to mean different things depending on the biological context, so we tried to be as clear as possible in our revision of that section.

Nonetheless, the reviewer is right that the use of OU models (over simpler models) merits consideration. Whether OU models can reasonably be applied to one's phylogenetic data is a highly relevant question because more complex models like OU can be incorrectly favored over simpler models if the statistical power of the analysis is weak^{14,15}. While our phylogenetic dataset is relatively large, it is still good practice to ensure that one's data are suitable for more complex model fitting procedures. To this end, we took a step further to address potential implications of our sampling by performing simulations to test the probability that our sampling might lead to type I error. We simulated trait evolution in two ways to be consistent with our OU analyses. We first simulated trait evolution 500 times on the MCC tree, and then simulated trait data once for each of the 500 trees from the posterior distribution. We simulated trait evolution via a BM process using the fastBM function in phytools¹⁶. We then fitted BM and OU models to the simulated data to determine what

percentage of analyses would incorrectly favor OU over BM. False positives were not an issue (0 %; Supplementary Table 2 and 3), supporting that we could reasonably fit OU models to our data. We incorporated this additional step into our revision.

Comment:

I'm not convinced by the conclusion in the Abstract (lines 40-43). If viviparity is not "...an adaptation to "cold climates per se," but instead "facilitates...colonization of cooler environments," then what favors the initial evolution of viviparity prior to that colonization? The classic "beauty" of the cold-climate hypothesis was that it allowed for intermediates states (egg retention in cold climates, leading eventually to viviparity). Thus, I don't see what favors the initial evolution of viviparity (intermediate steps) in a warm environment. Moreover, if a lower preferred body temperature is a key step in reducing metabolic rate in proto-viviparous species in a warm environment, then a clear disadvantage of that low T_p is that it must reduce potential activity time, at least in warm seasons. [See Sinervo et al. 2010.] So I must be missing something in the logic.

Response:

The reviewer makes a very important point about the interpretation of our results. Surely, viviparity allows females to regulate the incubation temperature of developing embryos, which impacts offspring performance and survivorship, and represents an advantage, as the reviewer notes, in cold habitats¹⁷⁻¹⁹. We clarify that the advantages of complete intrauterine incubation, combined with lower CT_{min} , T_b , T_{pref} , and CT_{max} , and the consequent lower mass-specific metabolic rate in viviparous species, likely allows them be successful in cool environments. Viviparity (and intermediate steps, like egg retention) not only protect the developing embryos against abiotic factors (such as humidity, oxygen levels, and temperature^{20,21}), but also provide maternal protection against biotic factors (such as infections, and predators)^{22,23}. In considering the reviewer's point, we scaled the language back a bit to better reflect our results and interpretation. Given our results, our data support the notion that viviparity evolves as a reproductive investment strategy that protects offspring mortality against myriad potential abiotic and biotic hazards. Certainly, the pressure posed by low temperatures in cold environments is one such pressure, but not the only one. Put differently, therefore, our interpretation is that the invasion of cold environments may be sufficient, but not obligatory, for viviparity to evolve.

The reviewer also brings up an interesting point about lower preferred temperatures in viviparous species, and how that might affect activity rates. In phrynosomatids, and other squamates, viviparous species clearly prefer lower temperatures than their oviparous counterparts^{24,25}. As the reviewer notes, lower preferred body temperatures make viviparous species quite vulnerable to warm temperatures, as rising temperatures increasingly force individuals to limit surface activity and retreat to avoid overheating (*i.e.*, more hours of restriction)^{24,26}. But, the risk under warming (for example, as quantified by Sinervo et al. (2010)) is a separate question from ours, which centers around the behavioral/physiological phenotypes associated with transitions to this reproductive mode.

Some viviparous phrynosomatids are quite widely distributed in warm environments^{27,28}, and maintain their lower field body temperatures through thermoregulatory behavior reflecting fine-scale sun/shade use. For example, *Sceloporus bulleri*, *S. macdougalli*, *S. prezygus*, *S. serrifer*, and *S. stejnegeri* inhabit environments where mean annual temperatures ranged from 24.2 °C to 27.9 °C, and maintain lower field body temperature through high thermoregulatory accuracy and effectiveness (see Supplementary Table 2). These species, however, necessarily depend on the presence of high vegetation cover (or thermal refuges, for example larger boulders of granitic rock, or sinkholes), which provide patches of sun and shade, allowing individuals to behaviorally thermoregulate, and until recently, maintain adequate periods of activity for support viable populations²⁹. Correspondingly, viviparous species in warm habitats appear to rely on fine-scale thermal heterogeneity for behavioral thermoregulation.

Comment:

One other issue is the use of field T_b to estimate metabolic rate. I see two problems here. First, T_b is apparently sampled from active lizards, but most lizards are active only a few hours per day. Thus, if you are trying to calculate metabolic rate, you really need time-series data on T_b , including T_b during inactive periods. I realize these won't be available for these species. This is one reason why I'd use a biophysical approach -- one could couple NicheMapR and a behavioral model to roughly estimate metabolic rate and energetic costs. At a minimum you could speculate on whether your conclusions would likely hold if you had hourly T_b data over the entire reproductive season. Second, I assume you plugged mean T_b and mean mass into these calculations. This will introduce a minor error, given the nonlinearity of $E \times T$ relationships and $E \times M$ relationships. See this Savage paper.

Savage, V. M. 2004. Improved approximations to scaling relationships for species, populations, and ecosystems across latitudinal and elevational gradients. *Journal of Theoretical Biology* 227:525-534.

Response:

Yes, this comment echoes one above and emphasizes that whether MR reflect activity or inactivity needs to be better spelled out. In our revision, we clarify that our measures are of T_b during activity. During these lizards' inactivity period (at night), we did not measure field body temperatures. However, we did measure operative temperatures at night, which afford a good approximation of core temperatures that individuals would exhibit during their inactivity period (as phrynosomatids are diurnal). So, to explore the reviewer's comment in greater detail, we used nighttime operative temperatures measured from the same sites where did our population sampling to model mass-corrected metabolic rate of inactive individuals. We then re-ran our OUwie analyses to see whether patterns observed during the day would also play out at night. Based on these data we found that, during the nighttime inactivity period, viviparous species exhibit a lower mass-corrected metabolic

rate than oviparous species, reflecting their disproportionate representation in cold environments.

We agree with the reviewer that mass-specific metabolic rate, estimated with mean T_b and mean mass may not adequately reflect energetic differences between parity modes. Because metabolic rate scales in relation $\frac{3}{4}$ with body mass, and increases exponentially with body temperature, estimation of mean metabolic rate, from mean body mass, and mean T_b for each species could underestimate or overestimate metabolic rate (“the fallacy of averages”)^{30,31}. Therefore, we also modelled temperature-corrected metabolic rate for each body mass measured in each individual of each species, and we modelled mass-corrected metabolic rate for each T_b measured in each individual of each species. Lastly, we estimated mean temperature-corrected MR, and mean mass-corrected MR for each species, and we re-ran our OUwie analyses. We found that oviparous and viviparous phrynosomatids exhibit a shared optimal temperature-corrected MR, but viviparous species exhibit a lower optimal mass-corrected MR, which support our previous results. Differences in metabolic rate among parity modes relates to the lower T_b in viviparous species. We incorporate all of these analyses into our revision.

Comment:

I encourage you to contrast your findings with that of Ma et al. (2018). They used a biophysical model and estimated that the total cost of development is lower for viviparous species than for oviparous embryos. They concluded that the main advantage of viviparity is that it shortens development time in cold environments. There's some overlap and discrepancies in perspectives between these two papers.

Response:

This was a great suggestion. According to Ma et al. (2018)¹⁹, viviparity shortens embryonic developmental time in cold habitats because incubation temperature is higher than the incubation temperature of nests. Our comparisons of field-active body temperatures and operative temperatures confirm this, as well. Furthermore, viviparous species exhibit a lower embryonic energy consumption than oviparous species, which support our hypothesis that viviparity represents a multidimensional phenotype of lower energy consumption. Additionally, the results of Ma et al. (2018)¹⁹ suggest that, in terms of embryonic development and energy consumption, viviparity also is advantageous in warm environments, which support the idea that viviparity could evolve in warm environments, and that their thermal and metabolic physiology can facilitate life in cold environments. We incorporate this study in our revision.

Comment:

Similarly, Beuchat and Vleck (1990) measured metabolism of gravid *Sceloporus jarrovi* (viviparous), and found that these metabolism of gravid females during pregnancy is higher than expected based on maternal body mass.

Response:

This is a great observation as metabolic rate in pregnant females is higher than expected on maternal body mass^{4,32}. Yet, Beuchat and Vleck (1990)³² found a lower preferred body temperature during pregnancy (pregnant females exhibit a preferred body temperature of 32°C, whereas non-pregnant lizards prefer body temperatures of 34.5°C) as an adjustment to accommodate the thermal requirements for embryogenesis. They estimated that the metabolic rate for pregnant females at 32°C is similar to metabolic rate for non-pregnant lizards at 34.5°C. Their findings are consistent with our results; pregnant females exhibit lower field-body and preferred body temperatures as a metabolic compensation for complete intrauterine embryogenesis.

Comment:

[As an aside, I'd encourage you to look at the doubly-labeled water literature. Ken Nagy measured field metabolic rates on many lizards (including phrynosomatids), but I don't recall whether he studied any viviparous species. If so, this would be an interesting comparison.]

Response:

We agree that the doubly-labeled water literature affords an interesting comparison. We download the data base of Nagy 2005³³ for 54 species of squamates (47 oviparous and 7 viviparous), which include data of eight oviparous phrynosomatid species and one viviparous phrynosomatid species. To explore data and to test for differences in mass-specific field metabolic rate among oviparous and viviparous squamates, we performed a Mann-Whitney rank sum test and we found that viviparous species tends to exhibit a lower mass-specific metabolic rate than oviparous species (0.1096 ± 0.07 *SD* vs 0.1549 ± 0.08 *SD*, respectively), but we did not find significant differences ($U=97$, $p=0.08$), as the statistical power of the analysis was weak.

Comment:

Again, I like the paper. It is long and complex, and some of it is well outside my expertise.

Response:

Thank you for your helpful review!

Comment:

Miscellaneous comments. Key ones are preceded by ***. [Note that some issues are eventually addressed in the Methods section.]

Response:

Thank you for your miscellaneous comments, were very useful to clarify ideas.

Comment:

26 Delete "strong"

Response:

Deleted.

Comment:

27 Rewrite sentence as syntax is off. "origins" literally refers here to life history pathways, but you want it to refer to innovation.

Response:

We changed that sentence.

Comment:

31 add "in" in front of mass-specific production

Response:

We changed that sentence.

Comment:

34 But won't these changes slow development, and thus the length of time the female is carrying embryos?

Response:

Incubation temperature influence embryonic development. At lower temperatures of incubation, embryogenesis is slow³⁴. For example, pregnant females of *Sceloporus jarrovii* maintained at constant temperature of 26°C exhibit a longer time of gestation, whereas, pregnant females maintained at 28, 30, and 32°C exhibit a similar length of gestation³⁵. Therefore, reductions in core-temperature (not exceeding 28°C) could decrease the mass-specific metabolic rate, without negative impacts in length of gestation. We clarify this point.

Comment:

35 What is a "thermal habitat"

Response:

By thermal habitat, we refer to thermal conditions in a given locality based on the corresponding bioclimatic variables (bio1, bio10, and bio11).

Comment:

38 "track" seems inappropriate, given a million-year lag!

Response:

Changed.

Comment:

39 Unclear.

Response:

Changed.

Comment:

41- Unclear.'

Response:

Changed.

Comment:

54 Squamates? or other taxa, too?

Response:

The mass-specific metabolic rate is lower in squamates³⁶ and in other viviparous taxa³⁷.

Comment:

96... Many traits here. Were the analyses also done with mean values, thus assuming no error in measurement?

Response:

Yes, analyses were performed using only mean values. To explore the effect of error in measurement, we re-ran SLOUCH analyses incorporating standard error, and we found the same evolutionary patterns.

Comment:

107 I assume you mean "mean annual AIR temperature" here. Given that some lizards have broad geographic ranges, how did you assign single temperatures to a species? Or did you use only the particular locality that you studied.

Response:

We included mean annual air temperature data for each locality representative of each species, where T_b , T_{pref} , CT_{min} , and/or CT_{max} were measured. We clarify the approach in the revised methods section. In Supplementary Data 2 we also specify the coordinates corresponding to the localities where thermal physiological data were measured.

Comment:

113 Syntax. "To...results, we used evolutionary regressions to explore..."

Response:

Changed.

Comment:

162 *** That is clear, but is age of first reproduction independent of body size? And is age of first reproduction independent of parity? Age of first reproduction is a dominant contributor to fitness (Cole 1954, Stearns, etc.)

Response:

In phrynosomatids there is a positive relationship between female body length and age at sexual maturity³⁸. Further, age at sexual maturity does not differ among parity modes³⁸.

Comment:

165 *** Personally, I doubt that there's global stabilizing selection on "quality and quantity of offspring." Clutch and egg size varies dramatically among populations in *Sceloporus occidentalis* and from clutch to clutch in *Uta stansburiana* (old work by Sinervo). Sinervo showed that optimal egg size varied among clutches, as I recall.

Response:

We agree with the comment, as the offspring size and number maximizing fitness can vary even among populations of the same species, or through of time, depending on different selective pressures. However, the general pattern is that offspring size and offspring number (per litter or clutch) are strongly correlated with female body size. Correspondingly, evolutionary changes in body size in the transition to live birth could impact simultaneously offspring size and/or offspring number.

Comment:

179 Rewrite. "..., females increase in mass but reduce Tb

Response:

Done.

Comment:

183 Here you use "optimal temperature" in a very different way from the rest of the text. This is confusing to me.

Response:

We clarified the use of optimal temperature throughout the text.

Comment:

Interesting idea, but the mass increases because of embryos, which will probably have high mass-specific rates because they are small and developing. The impact of this will be small, however.

Response:

This comment echoes an above, increase in body mass (related with embryos), and the embryonic metabolism could cause higher energetic demands in pregnant females³². However, decreases in thermal preferences of pregnant females could compensate the energetic cost of pregnancy³².

Comment:

187 Check Beuchat and Elner. Interesting idea.

Response:

This is a highly relevant paper. We cite it to support some ideas within our main text.

Comment:

***189 You are using instantaneous values for metabolic rate, but given that gestation time is much longer in viviparous species, what about cumulative metabolic rates (over entire reproductive period). You could do a quick analysis to check to see whether lower body temperature do compensate for longer gestations. I'm not convinced at a drop in T_b of a few degrees "compensates" for the much longer pregnancy.

Response:

Yes, we are using instantaneous values for metabolic rate, instead of cumulative metabolic rates. Yet, a reduction of 2.5°C in pregnant females of *S. jarrovii* could results in similar metabolic rates, compared with non-pregnant individuals³². We clarify this distinction in our revision, as the reviewer makes a key point.

Comment:

***196 Please explain your logic here. Are you referring to body or local temperature?

Response:

We refer to a strong evolutionary match between environmental temperatures and CT_{min} , T_b , T_{pref} , and/or CT_{max} .

Comment:

201 When you present results, you focus on "optimal" values not actual values. The latter are mor meaningful to me, perhaps because I'm not understanding optimal in this context.

Response:

We added additional detail in the methods to help clarify what is meant by optimal in our revision. We also included the actual values for each species within Supplementary Data 2.

Comment:

230 I don't follow.

Response:

We re-wrote this sentence.

Comment:

231 Shouldn't you use operative temperature here, as air temperature can be a misleading indicator of the thermal environment? See work by Porter, Kearney, Sunday, and many others.

Response:

We agree with the reviewer that operative temperatures could be a better indicator of thermal environment. We re-ran PGLS analyses with operative temperatures, and we found the same patterns, which supports our previous results. We added these new analyses to the revised manuscript.

Comment:

***Also, consider looking at seasonality (difference between hot and cold seasons). This will provide crude indication of the length of the active season (inverse relationship) I suspect.

Response:

Looking at seasonality was an excellent suggestion, as a high thermal variation could increase mortality and therefore natural selection favors high productivity³⁹. Compared with viviparous species, oviparous phrynosomatids inhabit environments with high seasonality²⁷ and exhibit high mass-specific production, which supports that oviparity and viviparity are on opposing ends of the fast-slow life history continuum. We include this in our revised manuscript.

Comment:

248 I realize this citation came from a reviewer, but the sentence is unclear to me. I could imagine many things that could be called fitness-related behaviors. If you are referring to the co-evolution of T_{pref} and T_{opt} , see Huey & Bennett Evolution 1987

Response:

We agree. Changed.

Comment:

Fig. 3 Interesting patterns. How do you account for the negative slope (though not significant) for viviparous lizards in 3c?

Response:

Gestation length is influenced by incubation temperature during activity and inactivity time. We think that higher $T_{b's}$ at activity time in cold habitats could accelerate embryonic development, to compensate for lower body temperatures at night.

Comment:

***In 3C, there are two outlier points: the left most oviparous, and the lowest viviparous. I was taught that comparisons of regressions should use shared ranges of the x variable, where possible. If you toss out the left-most MAT, which will have high leverage, is the correlation still significant? If you toss out the one viviparous species with $T_{pref} \sim 28^\circ\text{C}$, what happens to the correlation? I'm dubious about that low T_{pref} . None of the T_{pref} in table S1 are that low. Are you confident that is a valid estimate?

Response:

The left-most oviparous species corresponds to *Sceloporus graciosus*, and the lowest viviparous species corresponds to *S. macdougalli*. As the reviewer observed, the slope for oviparous species is significant, but is relatively weak. When we exclude *S. graciosus* within PGLS analyses, we did not find a correlation between MAT and T_{pref} (Supplementary Table 4, Fig 3c in the main text), whereas excluding (or including) *S. macdougalli* does not impact our general findings. We estimated T_{pref} data for *S. macdougalli* ($\sim 28^\circ\text{C}$) ourselves, so we are confident that the data are correct. In our revision, we excluded *S. graciosus* from this and the SLOUCH analyses.

Comment:

253, 254 $p = 0$? Report P values to more digits.

Response:

Changed.

Comment:

259 Something is wrong in this sentence.

Response:

Corrected.

Comment:

269 Operative temperatures would be more robust here.

Response:

As operative temperatures relate to the perches lizards actually use in the field, they would bias the results to their preferred microclimates. Our goal here was to ask whether, at a broad scale, transitions in thermal macrohabitat predict parity mode evolution, so the macroclimatic variables were used.

Comment:

***271 I think you are saying that viviparity evolves first and then this allows species to move into cooler environments. If so, what favors the evolution of viviparity? Am I missing something.

Response:

We clarify the idea, but yes, the reviewer describes the notion correctly. To be clear, temperature itself could facilitate the evolution of viviparity (per the CCH), but that the evolution of viviparity could reflect variables beyond temperature. We clarify this point throughout the manuscript.

Comment:

274 Doesn't the idea go back to Mell, or Weekes, or Sergeev?

Response:

We changed that sentence.

Comment:

296 I'm not convinced that viviparous females do have a reduced "mass-specific energetic burden." Yes, they will on an instantaneous basis, but you need time series of T_b (including inactive periods) and a factor for the much longer gestation. In other words, metabolic rate integrated over the entire reproductive period. Cumulative MR might still be lower, but that is unknown at present.

Response:

Yes, and this comment echoes a comment above. We clarify that we refer to energetic burden of allocation in maintenance and production, instead of pregnancy. Now, we model mass-corrected metabolic rate during both activity and inactivity periods, for each T_b measured in each individual of 65 species, and for each T_e recorded for each null model during the inactivity time of 42 species, respectively. With this approach we incorporate a time series of field body temperatures, and operative temperatures, encompassing full days. Our results support the previous findings: viviparous species exhibit a lower mass-corrected metabolic rate during periods of both activity and inactivity. Although we do not model metabolic rate for pregnant females, the lower preferred body temperature during pregnancy could match the energetic demands in non-pregnant individuals³².

Comment:

***357 Supplementary Data 2 is not in the "Supplementary information." I'd really like to see the input data, sample sizes, sources (literature vs. original). The only data on T_b , T_{pref} , etc. are for about $N \sim 26$ species (male vs female comparisons) in Supplementary Table 1.

Response:

Using the link below you could see the Supplementary Data 2.

<https://www.researchsquare.com/article/rs-127587/v1>

Comment:

362 "fails TO right"

Response:

Added.

Comment:

365 your new data collection? Sentence unclear

Response:

Added.

Comment:

367 Unclear on environmental variables. If a species was known from 100 localities, did you compute weighted values of mean temperature, for example, or from only the locality where Tb data were collected? I think the latter.

Response:

We agree this could be better stated. We now clarify our approach in the revised methods.

Comment:

372 See Beuchat & Ellner, below

Response:

We support some ideas in our paper with that citation.

Comment:

***373 Why exclude data on pregnant females, as aren't these the focus of your study? I suspect this is because the lab data aren't available. But if females drop Tp when pregnant, won't this just strengthen the pattern you observe? If so, you should consider saying that your estimates are thus conservative.

Response:

Yes, we excluded data on pregnant females because there are not sufficient data available. Where data are available, we included those into our revision to contextualize our results.

Comment:

375 Did you use paired t-tests?

Response:

No, we used unpaired t-tests as we were not making repeated comparisons for the same individuals.

Comment:

375 This is not universally the case. Compare Lailvaux (2007) vs Huey and Pianka (2007)

Response:

Yes, it is not a universal pattern in squamates, but is relative common in phrynosomatids, and our t-tests support similar values between sexes.

Comment:

379 Now we understand the decision in line 373. But are these data completely lacking?

Response:

Comparatively, there are relative few thermal physiological data of pregnant females, precluding robust evolutionary analyses.

Comment:

388 Ref 41 looked at 1 population, but the wording here implies multiple species were validated.

Response:

Yes, that citation corresponds with an example. We clarify this in the revision.

Comment:

393 1-5 days of sampling is hardly ideal, as Tb on those days might reflect weather or season, etc. These lizards "exhibit surface activity" for many months of the year, if not all year in some localities. How can a 1-5 day snapshot adequately characterize a species? [e.g., look at seasonal variation in Tb in Kalahari lizards, Huey and Pianka 1977.] Furthermore, how did you pick localities? Did you visit localities where physiological data were taken? You could use NicheMapR to get a more comprehensive portrait of the thermal environment.

Response:

Yes, we agree this could be more robust, but the sampling was nonetheless very laborious (encompassing, in total, many months of field work), and presents to date the most complete sampling of field-active behavior in phrynosomatid lizards (with new data for several dozen species). For newly collected data, we picked localities where focal species were abundant, and encompassing a wide latitudinal and altitudinal range. In addition, the field work took into account other variables, like the safety of workers, as several sites in Mexico where the species are found are currently experiencing unsafe conditions due to local turmoil (as a result of cartel influence). Therefore, our strategy was to get as much data as possible, from as many species as possible, while balancing effort, altitudinal and latitudinal variation, and ensuring researcher safety.

Comment:

395 Check Beuchat.

Response:

Thank you for the suggestion, we support our idea with citation of Beuchat.

Comment:

395 *** I assume that the studies here compared *active* body temperatures of viviparous vs. oviparous taxa. But lizards are active only for part of the day, and active T_b thus fail to capture the 24-h dynamics of T_b . Has anyone used telemetry on viviparous vs oviparous lizards in the same habitat?

Response:

Yes, body temperatures for these studies were obtained during the activity period of individuals. In our work we estimated operative temperatures at night (as an estimation of T_b at inactivity time), and we found that are lower for viviparous species ($U=168$, $p=0.1$; mean $T_{e-night}= 21^{\circ}\text{C}$ ($n=24$) for oviparous and 15.6°C ($n=24$) for viviparous species). We do not know of studies using telemetry for sympatric species in phrynosomatids.

Comment:

398 Wouldn't T_{pref} be the key metric here?

Response:

We clarify that corresponds with field body temperature.

Comment:

403... The results will vary seasonally in most species.

Response:

Yes, effectiveness in thermoregulatory behavior can vary seasonally. However, in our previous work, we documented that phrynosomatid species exhibit the same thermoregulatory pattern (that is, equally high thermoregulatory effectiveness) across seasons⁴⁰. Certainly, long-term data for all species would be even better, but these behavioral data provide a reasonable estimate of thermoregulatory patterns among species.

Comment:

420 Impressive. What fraction of these data are new versus taken from the literature?

Response:

Thank you. For this study we obtained new data of thermoregulatory effectiveness for 45 species, which represent ~70% of total data.

Comment:

427 As above, these are air temperatures.

Response:

Changed.

Comment:

438 What do you do with species with broad geographic ranges, which may have major variation in N eggs per clutch, N clutches per year, and size of eggs? (see Sinervo's old work with *Sceloporus*).

Response:

There is intraspecific variation in the life history traits we selected; correspondingly, for our sample size only one population representative of each species was selected. We describe this selection procedure in our methods.

Comment:

441 Some or perhaps many oviparous species have multiple clutches, but not all.

Response:

We clarify this in the revision.

Comment:

447 I'm surprised viviparous species don't live longer. How broad is the sampling in ref 45? You are missing one key element, namely the age of first reproduction. Given their often-cold habitats, viviparous species likely have delayed onset of reproduction.

Response:

In ref 45³⁸ data for longevity for 36 species were included. From these data, differences in age at sexual maturity among parity modes was not found³⁸.

Comment:

484 Why transform to integer value.

Response:

We did so to include the predicted mean body mass in the equation to estimate mass-specific metabolic rate.

Comment:

489 "neonate" or "egg" mass in oviparous species?

Response:

Neonate mass.

Comment:

534 Complex sentence. Perhaps start with "We performed PGLS...nlme to depict the evolutionary relationship between ..."

Response:

We agree. Changed.

Comment:

585 Move the citation for ref 13! I thought you were raising M to the 13th power!

Response:

Good point! Changed.

Comment:

Supplementary Information. I don't see a table with all of the data (field Tb, Tpref, size, etc. and the data sources.

Response:

We provide the data and data sources in the Supplementary Data 2 (<https://www.researchsquare.com/article/rs-127587/v1>).

Comment:

Beuchat, C. A. 1986. Reproductive influences on the thermoregulatory behavior of a live-bearing lizard. *Copeia* 1986:971-979.

Beuchat, C. A., and S. Ellner. 1987. A quantitative test of life history theory: thermoregulation by a viviparous lizard. *Ecological Monographs* 57:45-60.

Beuchat, C.A. and D. Vleck. 1990. Metabolic consequences of viviparity in a lizard, *Sceloporus jarrovi*. *Physiol. Zool.* 63:555-570.

Ma, L., Buckley, L.B., Huey, R.B., and Du, W.-G. 2018 A global test of the cold-climate hypothesis for the evolution of viviparity in squamate reptiles. *Global Ecol Biogr* 2018:1-11.

Ray Huey

Response:

Thank you so much for your helpful comments, Ray! We also included all these citations into our revision.

Reviewer 5

Comment:

There is a lot going on in this paper and in general there seems to be some striking differences in a range of characteristics between viviparous and oviparous lizards, which are pretty exciting. But the impact of the study is hard to judge at the moment because the presentation is complex and fragmented with predictions being only vaguely linked with hypotheses, which in some places don't seem to be articulated at all. The analyses are similarly hard to untangle, with a whole battery of tests being thrown at the data, but with no clear structure as to what the specific aim might be for each analysis set and how it relates to the overarching goals of the study. This could probably be remedied by having clearly argued hypotheses that then lead to a logical set of analyses that a reader can intuitively understand and relate back to those hypotheses. At the moment, however, the paper opens with a very general and very short 'background' paragraph and then gets quickly bogged down in the next paragraph with vague discussions of mass-specific metabolism and reproductive outputs that somehow lead to the set of predictions listed in Table 1. What the specific hypotheses might be that produce these predictions aren't really outlined, nor is it really clear how these predictions relate to the origin of viviparity briefly mentioned in the opening paragraph. Predation and life history are then mentioned at some point, too. The authors are heavily constrained here on the concise format of Nature Communications, but its even more vital for this format that a study is carefully structured to convey the study's goals and findings effectively.

Response:

We greatly appreciate the reviewer's feedback, as this encouraged us to deeply consider how we presented our study, both in the Introduction and the Results/Discussion. We took these comments to heart, as it is our hope to reach a broad audience. Correspondingly, a clear argument and structure is fundamental, especially since the life history, morphological, and thermal results require an integrative perspective for their proper contextualization. In our revision, we revisited and rewrote our manuscript with the reviewer's comments in mind. In our revised Introduction, we are clearer from the outset what our questions and goals are, and why those matter for understanding the evolution of viviparity. As the reviewer notes, there are several analyses that need to be unpacked. In our revision, we edited the Results/Discussion to provide a more cohesive structure, and built better transitions to more clearly guide the reader among the various results and interpretations. In addition to helpful feedback from Reviewer 3, we believe the edits we've made to the paper have resulted in a better articulated argument, with clearer flow and structure. We thank the reviewer for this accurate, and helpful, feedback.

Comment:

It seems to me that the study has two aims. The first is related to the title of the paper and is about quantifying the various characteristics that might have converged in viviparous species. The second seems to be about testing the putative adaptive origins of viviparity

itself. This isn't presented until very late in the paper, despite being mentioned near the start of the opening paragraph. And logically it seems that this 'why did viviparity evolve?' should start the study and then be followed by 'once it evolved, does it then lead to predictable adaptive changes in other characteristics?'

Response:

Yes, the reviewer makes an excellent point, and is one that relates to suggested analyses below. In our revision, we lead with this argument early on in the Introduction so as to better prime the reader for what is to come.

Comment:

The origin of viviparity seems to have been tied in the literature to the invasion of cold climates. All this information is presented in a single sentence in the opening paragraph and this probably needs to be outlined in more detail right at the outset so readers can follow. The authors have rightly taken the opportunity to test this idea in these lizards. The threshold analysis seems to be a relevant method for doing so, but there's no report of whether the effects computed by this analysis (I'm assuming the r values are effect sizes) are or are not statistically distinguishable from zero. Either these effects need to be reported with confidence intervals, or a model selection approach is needed to pit alternative models/outcomes against each other (like the BM vs OU models in the OUwie analyses). If the threshold analysis doesn't allow this, then a phylogenetic logistic regression would at least be able to provide this type of statistical benchmark. Here, if cold environments are associated with the shift to live birth, then you'd see the switch point in the trend line of the associated graph to the analysis. If it isn't — and it doesn't appear to be based on the reconstructions of temperature in the SI — then the analyses would be able to compute the effect (or lack thereof) and an associated confidence interval.

Response:

This was an excellent suggestion. To this end, we performed a phylogenetic logistic regression⁴¹ using the *phylolm* function in the *phylolm* package⁴² to test for linear relationships among parity mode and predictor traits (bio1, bio10, and bio11). We ran the analyses with 500 independent bootstrap replicates. We found that viviparity is not predicted by mean annual temperature ($z=-0.356$, $p=0.7$, $n=94$), mean temperature of the coldest quarter ($z=-0.0063$, $p=1$, $n=94$), or mean temperature of the warmer quarter ($z=-0.849$, $p=0.4$, $n=94$). These results parallel our findings using the threshold model, and we have incorporated this new analysis into the manuscript.

Comment:

Given cold environments don't seem to be associated with the origin of viviparity, are there no other hypotheses? It seems there should be an opportunity here to not only refute a prevailing hypothesis in the literature, but also to test other possible selection pressures for viviparity too. This would really push the paper up in impact.

Response:

In our revision we are clearer on how the cold climate hypothesis relates to the evolution of live birth in phrynosomatids. Viviparity is a strategy of parental care²³ that could be favored by thermal selection pressures, but also by other abiotic and biotic pressures (such as hypoxia, intra-and interspecific competition, and predation^{21,22,43}). We are clearer in the interpretation of our results in the revision. Our aim in this study was not test possible selection pressures associated with the evolution of viviparity. Rather, our aim was elucidate phenotypic convergences associated with evolution of live-bearing and to better contextualize the cold climate hypothesis with respect to transitions to viviparity. As such, we have re-written much of the manuscript to make this purpose clearer. As this comment also echoes those of Reviewer 3, we were clearer in the revision on the factors that shape transitions to viviparity.

Comment:

On the other aim of the study — the parallelisms highlighted in the title of the paper — I'm not sure the analyses applied by the authors really do justice to this. The optima shown in the figure of the main text is simply the phylogenetic mean of the characteristics in viviparous and oviparous species, but it doesn't capture the variance. Note, this isn't the variance in estimated optima across the 500 stochastic reconstructions of ancestor states (which is shown), but the variance of the individual optima computed for each of those reconstructions themselves. This is very different and goes directly to the extent these differences between viviparity and oviparity are in fact statistically distinguishable from one another. That is, these plots (or ones in the SI) need to be show the upper and lower confidence intervals of each of these computed optima. It can still be a jitter plot or a frequency distribution of points corresponding to the upper and lower CI values, but this way it will be clear what the statistical effect really is.

Response:

We agree with the reviewer. In our revision, we included CI values in Supplementary Table 2 and 3.

Comment:

Judging from the tables, it seems that OUwie is either identifying the multiple optima OU model as being the most supported or at least within 2 DAICs of an alliterative. But first, why are these single values? This has presumably been done 500 times across all the stochastic maps, and 500 times across a set of posterior trees, so why are these not mean AIC values or better yet, plotted as a frequency distribution so the overlap in support between the alternative models is clear. Second, why not just use the OU multiple peak model for all characteristics based on the fact its typically supported in most cases? It would then be possible to convert the phylogenetic half-life estimate into an alpha value (Hansen's original paper provides this equation) that provides a statistical measure of the rate of adaptation, as well as a sigma² value that measures the magnitude of stochasticity in the evolution process. These together would help show what I suspect is stabilising

selection operating on these characteristics in viviparity species, and perhaps not in the other oviparity species. The point being, focussing on just theta (the putative optima of the characteristic) provides a very small picture of what might have been happening in these lizards.

Response:

In our revision, we report in ST2 and ST3 the proportion of trees in which each model was best fit and the mean difference in AIC across the trees (Supplementary Table 2 and 3). Our motivation for running analyses separately (instead of using the OU multiple peak model for all characteristics) is two-fold. First, we did not know a priori whether it would be the case that the multi-peak model would be so robustly supported. Second, it actually isn't the best model for female body size or offspring size, and this finding (which differs from our results on the thermal traits) actually helps clarify some of the evolutionary patterns associated with transitions to live-birth. Likewise, from phylogenetic half-life we estimated alpha value (α). We found that $\alpha = \infty$ for CT_{min} in oviparous and viviparous lizards, which supports instantaneous adaptation to thermal environment. By contrast, α values for T_b , T_{pref} and CT_{max} were close to 0 in both parity modes, supporting very slow evolution in response to environmental temperature. We include alpha values within our Supplementary Table 5. This was a very helpful suggestion – thank you.

Comment:

More generally, it seems what the authors are really trying to do is document convergences among the five lineages of viviparity. The current analyses applied do not address this. The obvious approach would be to apply a phylogenetic PCA to identify how these characteristics are related to each other and which might be representative of orthogonal axes. And then use these representative characteristics or the PCA components themselves in a 'surface' analysis (the authors are probably aware of this R package already; if not, they can google it along with Ingram & Mahler and that should bring it up). This analysis will find where convergences have occurred across the entire phylogeny. Presumably these five lineages will be clearly identified as being convergent, but it will also reveal instances why they might not be... which would be interesting in itself. This would then be followed by something like Kevin Arbuckle's method (also available in R) to confirm the magnitude of convergences amongst viviparity species relative to the variance across the rest of the tree. Importantly this would provide confidence intervals as well, not just the magnitude of similarity.

Response:

Yes, we agree with the reviewer that one of our major goals is to understand which phenotypes tend to shift when viviparity evolves (and whether those shifts are robust to multiple origins of viviparity). A formal analysis of convergence strength, therefore, is an excellent suggestion. Following the reviewer's suggestion, we analyzed the strength of convergent evolution in morphology, thermal and metabolic physiology, and life history traits in viviparous phrynosomatids using the Wheatsheaf index (w)⁴⁴ in the windex

package in R⁴⁵. To estimate w it is necessary identify *a priori* convergent groups, which represent focal species for the analysis⁴⁴⁻⁴⁷. To this end, we used ancestral state reconstruction, which strongly supported five independent transitions to viviparity, consistent with previous results on this same group^{27,48}. Given the high type-I errors associated with SURFACE⁴⁹, we opted not use this method. Results of Wheatsheaf index found that, during the transition from oviparity to viviparity, there are strong convergences in thermal physiological traits (T_b , CT_{min} , and CT_{max}), mass-specific metabolic rate, annual fecundity, and mass-specific production (Supplementary Table 6). We incorporate this new analysis into our revision (it really did help make our argument clearer!) and thank the reviewer for their really helpful feedback.

Comment:

Currently this paper relies purely on a graphical presentation of theta and some basic model selection of BM, a common OU optima for all species and an OU that assumes viviparity and oviparity differ in optima. This latter analysis suggests convergence — or parrallism as the authors' put it — but it doesn't fully reveal it statistically. Again the OUwie analyse could still provide some resolution on changes in rates of adaptation to a viviparous lifestyle, and in particular the outcome of probable stabilising selection on these species (but not oviparous species).

Response:

Yes, this comment echoes the one above regarding convergence. As mentioned above, we performed the windex analysis to address questions related to phenotypic convergence among viviparous lineages.

Comment:

I really couldn't figure out the relevance of the mass-specific metabolism angle. Regardless, it's inferred metabolism not measured metabolism. So basically the authors have just applied a sophisticate transform of body size. If the authors are attempting to test the extent to which viviparous and oviparous species conform to the Gillooly-Brown metabolism equation, then ok, but how it relates to the overarching goals of the study is not obvious. Instead it seems to be a complex set of analyses that doesn't have a clear take home message or really relate back to the origin of viviparity (or its subsequent adaptive significance).

Response:

The reviewer brings up a good point related to what these MR can and cannot tell. As this comment is similar to several points raised by Reviewer 3, we provide detailed responses above. Briefly, the MR analysis (1) helped us show how energetic demand during activity and inactivity shifts between oviparous and viviparous lizards, and (2) helped us decipher the pathway (size versus thermal physiology) associated with those shifts. The windex analysis helped us documented the strength of convergence associated with this finding, as well.

Comment:

This is an exciting data set, and there might be something big in it, but it's really hard to extract what it is from the current paper. I suspect a shift in focus of the analyses would help tremendously in providing this resolution.

Response:

Thank you for these comments. We considered both reviewers' feedback seriously in restructuring our paper, with the intention of sharpening our argument and enhancing the structure and flow. We believe this has resulted in a stronger, clearer manuscript.

References

1. Gillooly, J. F., Brown, J. H., West, G. B., Savage, V. M. & Charnov, E. L. Effects of size and temperature on metabolic rate. *Science* **293**, 2248–2251 (2001).
2. Brown, J. H., Gillooly, J. F., Allen, A. P., Savage, V. M. & West, G. B. Toward a metabolic theory of ecology. *Ecology* **85**, 1771–1789 (2004).
3. Prieto, A. A. & Whitford, W. G. Physiological responses to temperature in the horned lizards, *Phrynosoma cornutum* and *Phrynosoma douglassii*. *Copeia* **1971**, 498–504 (1971).
4. Guillette, L. J. Effects of gravidity on the metabolism of the reproductively bimodal lizard, *Sceloporus aeneus*. *J. Exp. Zool.* **223**, 33–36 (1982).
5. Murrish, D. E. & Vance, V. J. Physiological responses to temperature acclimation in the lizard *Uta mearnsi*. *Comp. Biochem. Physiol.* **27**, 329–337 (1968).
6. Mayhew, W. W. Hibernation in the horned lizard, *Phrynosoma m'calli*. *Comp. Biochem. Physiol.* **16**, 103–119 (1965).
7. Mueller, C. F. Temperature and energy characteristics of the Sagebrush Lizard (*Sceloporus graciosus*) in Yellowstone National Park. *Copeia* **1969**, 153–160 (1969).
8. Dutton, R. H., Fitzpatrick, L. C. & Hughes, J. L. Energetics of the Rusty Lizard *Sceloporus olivaceus*. *Ecology* **56**, 1378–1387 (1975).
9. Beyer, E. C. & Spotila, J. R. Seasonal variation in metabolic rates and maintenance costs of the Eastern Fence Lizard, *Sceloporus undulatus*. *Comp. Biochem. Physiol. Part A Physiol.* **109**, 1039–1047 (1994).
10. Miles, D. B., Calsbeek, R. & Sinervo, B. Corticosterone, locomotor performance, and metabolism in Side-blotched Lizards (*Uta stansburiana*). *Horm. Behav.* **51**, 548–554 (2007).
11. Beaupre, S. J., Dunham, A. E. & Overall, K. L. Metabolism of a desert lizard: The effects of mass, sex, population of origin, temperature, time of day, and feeding on

- oxygen consumption of *Sceloporus merriami*. *Physiol. Zool.* **66**, 128–147 (1993).
12. DeMarco, V. Metabolic rates of female viviparous lizards (*Sceloporus jarrovi*) throughout the reproductive cycle: Do pregnant lizards adhere to standard allometry? *Physiol. Zool.* **66**, 166–180 (1993).
 13. Hansen, T. F., Pienaar, J. & Orzack, S. H. A comparative method for studying adaptation to a randomly evolving environment. *Evolution* **62**, 1965–1977 (2008).
 14. Ho, L. S. T. & Ané, C. Intrinsic inference difficulties for trait evolution with Ornstein-Uhlenbeck models. *Methods Ecol. Evol.* **5**, 1133–1146 (2014).
 15. Cooper, N., Thomas, G. H., Venditti, C., Meade, A. & Freckleton, R. P. A cautionary note on the use of Ornstein-Uhlenbeck models in macroevolutionary studies. *Biol. J. Linn. Soc.* **118**, 64–77 (2016).
 16. Revell, L. J. phytools: An R package for phylogenetic comparative biology (and other things). *Methods Ecol. Evol.* **3**, 217–223 (2012).
 17. Shine, R. A new hypothesis for the evolution of viviparity in reptiles. *Am. Nat.* **145**, 809–823 (1995).
 18. Shine, R. Evolution of an evolutionary hypothesis: A history of changing ideas about the adaptive significance of viviparity in reptiles. *J. Herpetol.* **48**, 147–161 (2014).
 19. Ma, L., Buckley, L. B., Huey, R. B. & Wei-Guo, D. A global test of the cold-climate hypothesis for the evolution of viviparity of squamate reptiles. *Glob. Ecol. Biogeogr.* 1–11 (2018).
 20. Andrews, R. M. Evolution of viviparity in squamate reptiles (*Sceloporus* spp.): A variant of the cold-climate model. *J. Zool.* **250**, 243–253 (2000).
 21. Watson, C. M. & Cox, C. L. Elevation, oxygen, and the origins of viviparity. *J. Exp. Zool. B Mol. Dev. Evol.* **336**, 457–469 (2021).
 22. Wourms, J. P. & Lombardi, J. Reflections on the evolution of piscine viviparity. *Am. Zool.* **32**, 276–293 (1992).
 23. Furness, A. I. & Capellini, I. The evolution of parental care diversity in amphibians. *Nat. Commun.* **10**, 1–12 (2019).
 24. Sinervo, B. *et al.* Erosion of lizard diversity by climate change and altered thermal niches. *Science* **328**, 894–899 (2010).
 25. Medina, M. *et al.* Thermal biology of genus *Liolaemus*: A phylogenetic approach reveals advantages of the genus to survive climate change. *J. Therm. Biol.* **37**, 579–586 (2012).
 26. Sinervo, B., Miles, D. B., Martínez-Méndez, N., Lara-Redendiz, R. & Méndez-De la Cruz, F. R. Response to comment on "Erosion of lizard diversity by climate change and altered thermal niches". *Science* **332**, 537b (2011).
 27. Lambert, S. M. & Wiens, J. J. Evolution of viviparity: A phylogenetic test of the

- cold-climate hypothesis in phrynosomatid lizards. *Evolution* **67**, 2614–2630 (2013).
28. Martínez-Méndez, N., Mejía, O., Ortega, J. & Méndez-de la Cruz, F. Climatic niche evolution in the viviparous *Sceloporus torquatus* group (Squamata: Phrynosomatidae). *PeerJ* **6**, e6192 (2019).
 29. Martínez-Méndez, N., Mejía, O. & Méndez-de la Cruz, F. R. The past, present and future of a lizard: The phylogeography and extinction risk of *Sceloporus serrifer* (Squamata: Phrynosomatidae) under a global warming scenario. *Zool. Anz.* **254**, 86–98 (2015).
 30. Savage, V. M. Improved approximations to scaling relationships for species, populations, and ecosystems across latitudinal and elevational gradients. *J. Theor. Biol.* **227**, 525–534 (2004).
 31. Dillon, M. E., Wang, G. & Huey, R. B. Global metabolic impacts of recent climate warming. *Nature* **467**, 704–706 (2010).
 32. Beuchat, C. A. & Vleck, D. Metabolic consequences of viviparity in a lizard, *Sceloporus jarrovi*. *Physiol. Zool.* **63**, 555–570 (1990).
 33. Nagy, K. A. Field metabolic rate and body size. *J. Exp. Biol.* **208**, 1621–1625 (2005).
 34. Qualls, C. P. & Andrews, R. M. Cold climates and the evolution of viviparity in reptiles: Cold incubation temperatures produce poor-quality offspring in the lizard, *Sceloporus virgatus*. *Biol. J. Linn. Soc.* **67**, 353–376 (1999).
 35. Beuchat, C. A. Temperature effects during gestation in a viviparous lizard. *J. Therm. Biol.* **13**, 135–142 (1988).
 36. Zhang, L., Guo, K., Zhang, G. Z., Lin, L. H. & Ji, X. Evolutionary transitions in body plan and reproductive mode alter maintenance metabolism in squamates. *BMC Evol. Biol.* **18**, 45 (2018).
 37. Healy, K., Ezard, T. H. G., Jones, O. R., Salguero-Gómez, R. & Buckley, Y. M. Animal life history is shaped by the pace of life and the distribution of age-specific mortality and reproduction. *Nat. Ecol. Evol.* **3**, 1217–1224 (2019).
 38. Zúñiga-Vega, J. J., Fuentes-G., J. A., Ossip-Drahos, A. G. & Martins, E. P. Repeated evolution of viviparity in phrynosomatid lizards constrained interspecific diversification in some life-history traits. *Biol. Lett.* **12**, 20160653 (2016).
 39. Pianka, E. R. On r- and K-Selection. *Am. Nat.* **104**, 592–597 (1970).
 40. Domínguez-Guerrero, S. F. *et al.* Interactions between thermoregulatory behavior and physiological acclimatization in a wild lizard population. *J. Therm. Biol.* **79**, 135–143 (2019).
 41. Ives, A. R. & Garland, T. Phylogenetic logistic regression for binary dependent variables. *Syst. Biol.* **59**, 9–26 (2010).
 42. Tung Ho, L. S. & Ané, C. A linear-time algorithm for gaussian and non-gaussian

trait evolution models. *Syst. Biol.* **63**, 397–408 (2014).

43. Stearns, S. C. Life-history tactics: A review of the ideas. *Q. Rev. Biol.* **51**, 3–47 (1976).
44. Arbuckle, K., Bennett, C. M. & Speed, M. P. A simple measure of the strength of convergent evolution. *Methods Ecol. Evol.* **5**, 685–693 (2014).
45. Arbuckle, K. & Minter, A. windex: Analyzing convergent evolution using the Wheatsheaf index in R. *Evol. Bioinforma.* 11–14 (2015).
46. Esquerré, D. & Keogh, J. S. Parallel selective pressures drive convergent diversification of phenotypes in pythons and boas. *Ecol. Lett.* **19**, 800–809 (2016).
47. Feilich, K. L. Correlated evolution of body and fin morphology in the cichlid fishes. *Evolution* **70**, 2247–2267 (2016).
48. Hodges, W. L. Evolution of viviparity in horned lizards (*Phrynosoma*): Testing the cold-climate hypothesis. *J. Evol. Biol.* **17**, 1230–1237 (2004).
49. Adams, D. C. & Collyer, M. L. Multivariate phylogenetic comparative methods: Evaluations, comparisons, and recommendations. *Syst. Biol.* **67**, 14–31 (2018).

Reviewers' Comments:

Reviewer #3:

Remarks to the Author:

I reviewed the responses to my comments and (briefly) of those of the other reviewer. I think the authors have done a fine job of responding to criticisms and suggestions, and they have added additional analyses that directly address suggestions.

I liked the original ms. and (not surprisingly) I like this revision. I recommend publication following minor revision. This paper tackles a topic that has been around for many decades and adds new and interesting perspectives.

In my review, I noted that they really needed to have 24h-h Tb profiles, as inactive temperatures can be very different from active ones. The authors now use operative temperatures at night, which "afford a good approximation of core temperatures that individuals would exhibit during their inactivity period..." However, that is true only in the Te models were in the microhabitats used by these lizards when inactive. That isn't specified in the Methods (428..., 555). I doubt that active and inactive microhabitats are the same for these species, as phrynosomatid lizards (in my experience) retreat underground, under litter, or under rocks when inactive, where Te can be very different from those where the lizards are active by day. In any case their use of Te is a good first step in this direction, but I encourage them to concede that their nocturnal Te may be different from those experienced by inactive lizards.

The text is usually clear, but I think the Abstract in particular needs work.

line 30-33 complex and hard to follow. rewrite.

line 33 "prevalent" means of species or disproportionately more common than oviparous species?

line 34 are "transition in thermal environment" bi- or uni-directional?

line 38 convergent physiological shifts -- and behavioral as well?

line 42 should you introduce the cold climate hypothesis near the beginning of the abstract.

line 48 "prolific" ?

line 67 unclear

line 73 more clearly define "mass-specific production" (following Meiri et al.) as the mass of offspring produced yearly divided by female mass

line 77 rewrite

can't having fewer clutches/year contribute to mass-specific production? (not in Table 1)

line 148 I realize you are citing a reference, but wouldn't temperature seasonality be greater at high elevation than low?

I like Fig. 1

line 234 Given the temperature-size rule, wouldn't eggs at low temperatures result in larger offspring?

line 296 "shifts in the thermal environment are not strongly associated with evolutionary transitions to viviparity."

I think the traditional idea is that a shift in the thermal environment (warm to cold) favors an evolutionary transition to viviparity. so perhaps re-order 'thermal environment' and 'viviparity' in the above sentence?

line 335 "their over-representation" literally refers to behavioral and physiological properties, which is not what you mean.

incidentally, use British spelling.

line 442 I prefer "have lower field body temperatures" to "exhibit lower..."

Supplementary Table 1. In general I prefer over-under comparisons (female vs. male Tb) over side-by-side comparisons, as is used here. Over-under comparisons are much easier, and every column represents a single kind of data.

You do over-under in other tables.

Mean Tb (N)

Callisaurus draconoides females 39 ± 0.53 16

males 38.99 ± 0.73 12

Ray Huey

Reviewer #3

Comment:

I reviewed the responses to my comments and (briefly) of those of the other reviewer. I think the authors have done a fine job of responding to criticisms and suggestions, and they have added additional analyses that directly address suggestions. I liked the original ms. and (not surprisingly) I like this revision. I recommend publication following minor revision. This paper tackles a topic that has been around for many decades and adds new and interesting perspectives.

Response:

Thank you so much for your kind words. We appreciated all the feedback from review, and incorporating your past and new suggestions has resulted in a stronger manuscript.

Comment:

In my review, I noted that that they really needed to have 24h-h Tb profiles, as inactive temperatures can be very different from active ones. The authors now use operative temperatures at night, which "afford a good approximation of core temperatures that individuals would exhibit during their inactivity period..." However, that is true only in the Te models were in the microhabitats used by these lizards when inactive. That isn't specified in the Methods (428..., 555). I doubt that active and inactive microhabitats are the same for these species, as phrynosomatid lizards (in my experience) retreat underground, under litter, or under rocks when inactive, where Te can be very different from those where the lizards are active by day. In any case their use of Te is a good first step in this direction, but I encourage them to concede that their nocturnal Te may be different from those experienced by inactive lizards.

Response:

That was an excellent observation. We edited the Methods section to denote that operative temperatures at night were measured in microsites where lizards were active during the day, and that may be different than microenvironmental temperatures experienced by inactive lizards in their retreats. Nonetheless, prior work on a group of montane, saxicolous anoles¹ showed that nocturnal retreat sites (*e.g.*, below boulders and in rock crevices) differed surprisingly little in operative temperature (~1C). So, the measures are likely to be a fairly close (though imperfect) approximation. We clarify this in our revision.

¹Muñoz M. M. and Losos J. B. Thermoregulatory behavior simultaneously promotes and forestalls evolution in a tropical lizard. *Am. Nat.* 191, E15-E26 (2018).

Comment:

The text is usually clear, but I think the Abstract in particular needs work.

Response:

Following the clear suggestions below, we modified the abstract accordingly, and made it clearer.

Comment:

line 30-33 complex and hard to follow. rewrite.

Response:

Changed.

Comment:

line 33 "prevalent" means of species or disproportionately more common than oviparous species?

Response:

We changed that sentence.

Comment:

line 34 are "transition in thermal environment" bi- or uni-directional?

Response:

Good point! In phrynosomatids, transitions are unidirectional, because viviparous species (from cold habitats) evolved from oviparous species (from warm habitats). We clarify the sentence.

Comment:

line 38 convergent physiological shifts -- and behavioral as well?

Response:

Right. We added this to the sentence

Comment:

line 42 should you introduce the cold climate hypothesis near the beginning of the abstract.

Response:

Added.

Comment:

line 48 "prolific" ?

Response:

Changed

Comment:

line 67 unclear

Response:

We cut out that sentence, because this was explained in the previous paragraph.

Comment:

line 73 more clearly define "mass-specific production" (following Meiri et al.) as the mass of offspring produced yearly divided by female mass

Response:

Changed.

Comment:

line 77 rewrite

Response:

Changed.

Comment:

can't having fewer clutches/year contribute to mass-specific production? (not in Table 1)

Response:

Of course, the number of clutches per year contribute to mass-specific production. Yet, our trait combinations are structured considering transitions from oviparity to viviparity. Therefore, we consider only hatchlings/year into trait combinations.

Comment:

line 148 I realize you are citing a reference, but wouldn't temperature seasonality be greater at high elevation than low?

Response:

It is a good observation, and it is results from the geographic distribution of phrynosomatids. Viviparous phrynosomatids are more common in tropical habitats (where there is low temperature seasonality), whereas oviparous phrynosomatids are more common in temperate environments (where there is high temperature seasonality)².

²Lambert, S. M. & Wiens, J. J. Evolution of viviparity: A phylogenetic test of the cold-climate hypothesis in phrynosomatid lizards. *Evolution* 67, 2614–2630 (2013).

Comment:

I like Fig. 1

Response: Thank you!

Comment:

line 234 Given the temperature-size rule, wouldn't eggs at low temperatures result in larger offspring?

Response:

Yes, typically individuals exposed to lower temperatures during their development tend to be larger. However, empirical approaches with phrynosomatids have shown that eggs incubated at colder temperatures tend to produce small neonates^{3,4}. Likely, this is associated with exceeding the optimal temperature range for embryogenesis. For example, in the viviparous *Sceloporus jarrovii*, neonates tend to be smaller if embryogenesis occurs at 26°C or 36 °C, as compared with neonates incubated at ~30°C⁵.

³Qualls, C. P. & Andrews, R. M. Cold climates and the evolution of viviparity in reptiles: Cold incubation temperatures produce poor-quality offspring in the lizard, *Sceloporus virgatus*. *Biol. J. Linn. Soc.* 67, 353–376 (1999).

⁴Andrews, R. M., Mathies, T. & Warner, D. A. Effect of incubation temperature on morphology, growth, and survival of juvenile *Sceloporus undulatus*. *Herpetol. Monogr.* 14, 431 (2000).

⁵Beuchat, C. A. Temperature effects during gestation in a viviparous lizard. *J. Therm. Biol.* 13, 135–142 (1988).

Comment:

line 296 "shifts in the thermal environment are not strongly associated with evolutionary transitions to viviparity."

Response:

Changed.

Comment:

I think the traditional idea is that a shift in the thermal environment (warm to cold) favors an evolutionary transition to viviparity. so perhaps re-order 'thermal environment' and 'viviparity' in the above sentence?

Response:

Changed.

Comment:

line 335 "their over-representation" literally refers to behavioral and physiological properties, which is not what you mean. incidentally, use British spelling.

Response:

Changed.

Comment:

line 442 I prefer "have lower field body temperatures" to "exhibit lower..."

Response:

Changed.

Comment:

Supplementary Table 1. In general I prefer over-under comparisons (female vs. male Tb) over side-by-side comparisons, as is used here. Over-under comparisons are much easier, and every column represents a single kind of data.

You do over-under in other tables.

Mean Tb (N)

Callisaurus draconoides females 39 ± 0.53 16

males 38.99 ± 0.73 12

Response:

Changed.

Ray Huey